# Acetylation of histone H3 at lysine 64 regulates nucleosome dynamics and facilitates transcription

Vincenzo Di Cerbo[1,2], Fabio Mohn[3†], Daniel P Ryan[4†‡], Emilie Montellier[5,6], Salim Kacem[7], Philipp Tropberger[1,2], Eleni Kallis[8], Monika Holzner[8], Leslie Hoerner[9], Angelika Feldmann[9], Florian Martin Richter[10], Andrew J Bannister[11,12], Gerhard Mittler[13], Jens Michaelis[8], Saadi Khochbin[5,6], Robert Feil[7], Dirk Schuebeler[9], Tom Owen-Hughes[4], Sylvain Daujat[1]*, Robert Schneider[1]*

[1]Department of Functional Genomics, Institut de Génétique et de Biologie Moléculaire et Cellulaire (IGBMC), CNRS UMR, Strasbourg, France; [2]Max Planck Institute of Immunobiology and Epigenetics, Freiburg, Germany; [3]Institute of Molecular Biotechnology, Vienna, Austria; [4]Wellcome Trust Centre for Gene Regulation and Expression, College of Life Sciences, University of Dundee, Dundee, United Kingdom; [5]INSERM U823, Université Joseph Fourier, Grenoble, France; [6]Faculté de Médecine, Institut Albert Bonniot, Grenoble, France; [7]Institut de Génétique Moléculaire, CNRS UMR5535/Université de Montpellier I and II, Montpellier, France; [8]Institute for Biophysics, Ulm University, Ulm, Germany; [9]Friedrich Miescher Institute for Biomedical Research (FMI), Basel, Switzerland; [10]Cellular Immunobiology, Max Planck Institute of Immunobiology and Epigenetics, Freiburg, Germany; [11]Gurdon Institute, Cambridge, United Kingdom; [12]Department of Pathology, University of Cambridge, Cambridge, United Kingdom; [13]Max Planck Institute of Immunobiology and Epigenetics, Freiburg, Germany

*For correspondence: daujat@igbmc.fr (SD); schneidr@igbmc.fr (RS)

†These authors contributed equally to this work

Present address: ‡The John Curtin School of Medical Research, The Australian National University, Canberra, Australia

Competing interests: The authors declare that no competing interests exist.

**Abstract** Post-translational modifications of proteins have emerged as a major mechanism for regulating gene expression. However, our understanding of how histone modifications directly affect chromatin function remains limited. In this study, we investigate acetylation of histone H3 at lysine 64 (H3K64ac), a previously uncharacterized acetylation on the lateral surface of the histone octamer. We show that H3K64ac regulates nucleosome stability and facilitates nucleosome eviction and hence gene expression in vivo. In line with this, we demonstrate that H3K64ac is enriched in vivo at the transcriptional start sites of active genes and it defines transcriptionally active chromatin. Moreover, we find that the p300 co-activator acetylates H3K64, and consistent with a transcriptional activation function, H3K64ac opposes its repressive counterpart H3K64me3. Our findings reveal an important role for a histone modification within the nucleosome core as a regulator of chromatin function and they demonstrate that lateral surface modifications can define functionally opposing chromatin states.

## Introduction

Histone modifications are central to the regulation of all chromatin-based processes. Four core histones—H3, H4, H2A, and H2B—comprise the nucleosomal core particle, and each may be decorated with multiple covalent modifications, including acetylation, methylation, phosphorylation, sumoylation, and ubiquitination (*Kouzarides, 2007*). To date, most attention has focused on modifications within the

**eLife digest** DNA is a very long molecule, so it needs to be packaged carefully to fit into the nucleus of a cell. To achieve this, the DNA is wrapped around proteins called histones to form a structure termed a nucleosome, which is the building block of a more compacted substance called chromatin. However, to express the genes in the DNA it is necessary to open up parts of the chromatin to give various enzymes access to the DNA.

Cells often chemically modify histones by adding acetyl or methyl groups, and these modifications are known to influence what proteins can bind to the nucleosomes, which ultimately influences what genes are expressed in the cell at a given time. It has been suspected for some time that histone modifications can also influence gene expression more directly, but there has been little evidence for this idea.

Now Di Cerbo et al. have studied what happens when acetyl or methyl groups are added to a specific site within a histone called H3K64, which is close to where the DNA wraps around this histone. These experiments showed that this site tends to be acetylated when a nearby gene is active, and to be unmodified or methylated when this gene is not active. It appears that the addition of the acetyl group makes this region of the chromatin less stable: this, in turn, makes it easier for the chromatin to be unpacked, thus giving access to the enzymes that transcribe the DNA and allowing transcription to take place. The work of Di Cerbo et al. shows that methylation and acetylation at the same site within a histone can define two opposing states of chromatin and DNA: an active state and a repressive state.

flexible N-terminal tails of histones, which extend from their own nucleosome. Due to their accessibility, 'reader' or effector proteins selectively bind to modified sites in the tails to mediate downstream effects. In this way 'readers' can provide a relatively simple mechanism enabling cells to decipher the so-called 'histone-code', to facilitate the regulation of biological processes such as transcription, DNA replication, and damage repair.

Interestingly, covalent modifications also occur within the globular domain of histones (*Garcia, 2009*), especially at positions that are in close contact with the nucleosomal DNA wrapped around each octamer. In particular, modifications on the outer surface of the histone octamer, the so-called lateral surface, have the potential to directly influence chromatin structure by altering histone–histone or histone–DNA interactions (*Cosgrove, 2007*; *Tropberger and Schneider, 2010*; *Tropberger et al., 2013*). Because of their structurally important position close to the DNA, one can directly address the mechanism(s) by which these lateral surface modifications impact on nucleosome dynamics and chromatin function. This is in contrast to tail modifications that play a mainly indirect role in chromatin regulation through recruitment of effector proteins.

Understanding how histone modifications ultimately impact on chromatin function is still an open challenge and this has not been helped by the fact that the repertoire of known modifications is far from complete (*Tan et al., 2011*). Here, we have investigated the functional mechanism of an uncharacterized acetylation site, lysine 64 of histone H3 (H3K64ac), which lies within the H3 globular domain. We demonstrate that this lateral surface modification can directly influence nucleosomal stability and dynamics, which consequently affects transcriptional regulation.

## Results

### H3K64ac is a novel histone modification enriched in euchromatin

We used mass spectrometry to identify novel histone modifications and found an uncharacterized acetylation site on histone H3 lysine 64 (H3K64ac) (*Figure 1A*, *Figure 1—figure supplement 1A,B*). H3K64 is the first amino acid of the H3 alpha1 helix, the first of three alpha helices in the histone fold. It is found on the lateral surface of the histone octamer in close proximity to the inner gyre of DNA (*Davey et al., 2002*; *Figure 1B*) at a location distinct from other potentially acetylated residues, such as H3K56.

To study the biological function of H3K64ac, we first performed an exhaustive characterization of an antibody raised against this modification. This antibody specifically detects endogenous acetylated

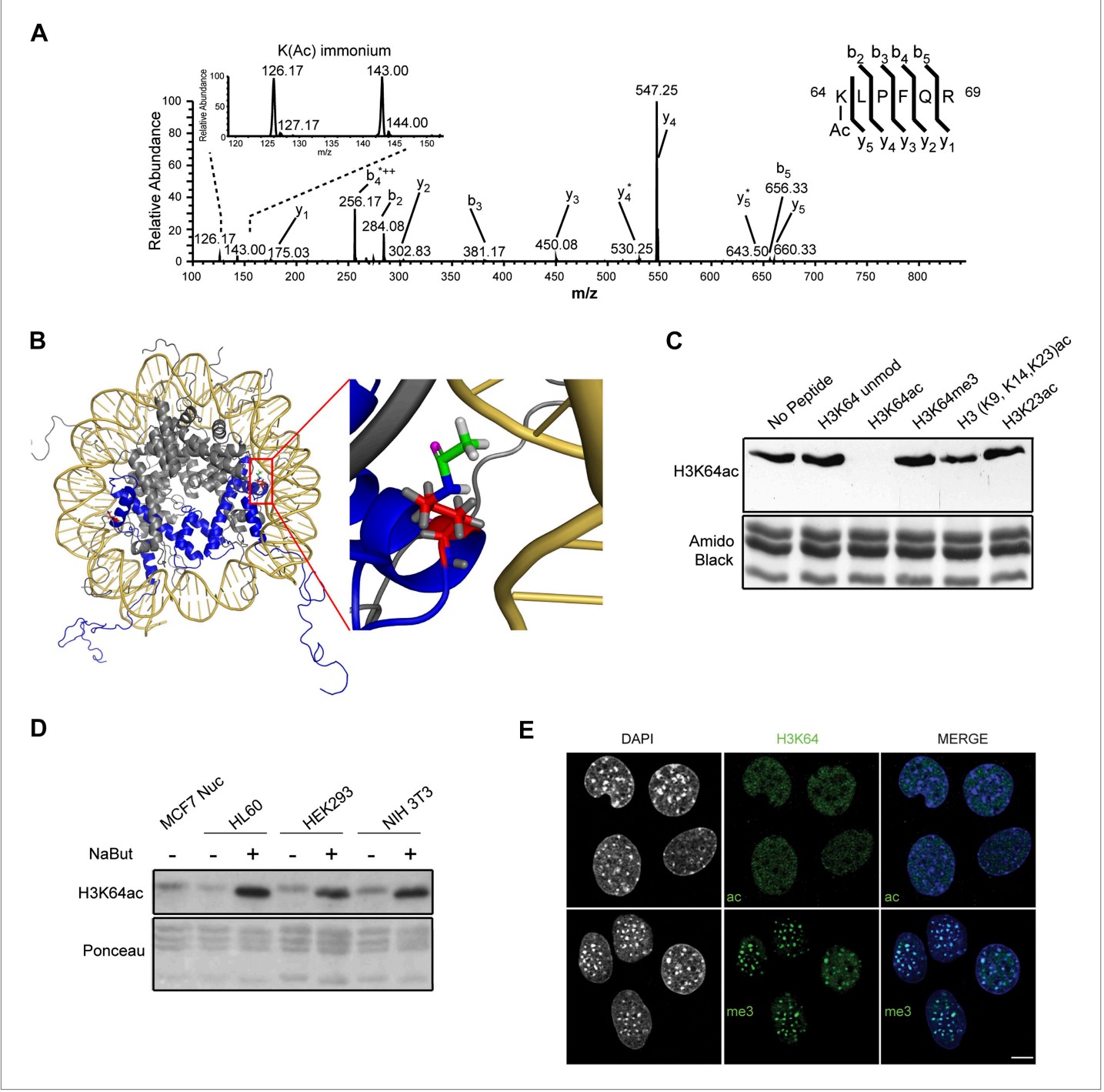

**Figure 1**. Acetylation of K64 in histone H3 is a novel histone modification. (**A**) CID MS/MS spectrum of the tryptic peptide (K(ac)LPFQR; m/z [MH$_2$$^{2+}$] 415.74823) derived from endogenous histone H3 demonstrating K64 acetylation. The presence of the b2, the y5, the y5-NH3(y5*) and the immonium ions derived from (ε) acetyl-lysine (enlarged spectrum inlet) are used for site localization to lysine 64. (**B**) H3K64 is on the lateral surface of the histone octamer. 3D modelling of a nucleosome. H3 dimer is shown in blue and the proximity of acetylated H3K64 (red rectangle) with the DNA is highlighted in the zoomed-in inset. In red are shown the main-chain and the side-chain of lysine 64. The acetyl group linked to the side-chain terminal nitrogen atom (in blue) appears in green, with its oxygen atom in purple. All hydrogen atoms are displayed in grey. (**C**) Peptide competition of H3K64ac immunoblot. H3K64ac antibody was pre-adsorbed with 50 pmoles/ml of indicated peptides. Amido black staining is shown as loading control (bottom panel). (**D**) Immunoblot analysis of H3K64 acetylation state in different cell lines, treated (+) or not (−) with HDACs inhibitor Na-Butyrate. Untreated MCF7 nuclesosomes were used as control. Ponceau staining is shown as loading control (bottom panel). (**E**) H3K64ac is enriched in euchromatic regions. H3K64

*Figure 1. Continued on next page*

*Figure 1. Continued*

acetylation (ac, upper panels) and tri-methylation patterns (me3, lower panels) in MEFs. DAPI dense foci represent pericentric heterochromatin. Single sections are shown. Scale bar, 10 µm.

The following figure supplements are available for figure 1:

**Figure supplement 1**. MS/MS spectra of H3K64ac peptides.

**Figure supplement 2**. Anti-H3K64ac antibody validation.

**Figure supplement 3**. H3K64ac is excluded from heterochromatic foci.

histone H3 (*Figure 1—figure supplement 2A*) and recognizes a K64-acetylated peptide with a high degree of specificity (*Figure 1—figure supplement 2B–E*) compared to other H3 acetylated lysines. Its recognition of H3 was efficiently competed by the immunizing peptide, but not by other peptides containing acetylated, methylated, or unmodified histone regions (*Figure 1C*, *Figure 1—figure supplement 2F*). Furthermore, limited tryptic digestion of native nucleosomes, which removes the H3 tails whilst leaving the DNA-protected H3 core region largely intact, confirmed the antibody's specificity. Indeed, this digestion treatment resulted in the loss of signal for the tail modifications, such as H3K9ac, H3K18ac, and H3K27ac. In contrast, the truncated H3 core was still recognized by the H3K64ac antibody at levels comparable to the undigested H3 (*Figure 1—figure supplement 2G*).

Using this antibody, we found that H3K64ac is present in a variety of mouse and human cell lines and tissues suggesting a rather ubiquitous function (*Figure 1D*, *Figure 1—figure supplement 2I*). Upon HDAC-inhibitor treatment, H3K64 acetylation levels increased (*Figure 1D*, *Figure 1—figure supplement 2I*). Immunofluorescence (IF) showed a distinct nuclear localization of H3K64ac with a relative depletion from heterochromatin (*Figure 1E*, *Figure 1—figure supplement 3*, compare panels 5 and 6, and *Figure 1—figure supplement 2H*). Interestingly, this localization pattern is the opposite to that of H3K64me3 (*Figure 1E*). We previously established that H3K64me3 is a novel repressive mark enriched in pericentromeric heterochromatin that might help to 'lock' the conformation and/or position of the nucleosome, and consequently the surrounding chromatin (*Daujat et al., 2009*; *Lange et al., 2013*)

## H3K64ac associates with active regulatory genomic regions

To obtain a comprehensive picture of the genomic distribution of H3K64ac, we performed ChIP-on-chip assays using chromatin isolated from mouse embryonic stem (ES) cells and Nimblegen tiling microarrays (*Figure 2—figure supplement 1A*; *Lienert et al., 2011*). In line with the euchromatic localization detected in IF, we found strong enrichment of H3K64ac at the transcriptional start site (TSS) of active genes (*Figure 2A*, *Figure 2—figure supplement 1A–C*). At TSSs, we detected a strong correlation between the enrichment of H3K64ac and RNA Polymerase II occupancy (*Figure 2B*), as well as with the presence of active histone marks (*Figure 2—figure supplement 1D*), suggesting a role for H3K64ac in transcriptional activation. Of note, local H3K64ac enrichment is indicative of the steady-state mRNA level of the respective gene (*Figure 2C*) and it is not simply due to increased histone H3 density (*Figure 2—figure supplement 1E*). Consistent with this, H3K64ac enrichment is anti-correlated with repressive marks such as H3K27me3 and H3K64me3 (*Figure 2—figure supplement 1D*). Furthermore, we find H3K64ac strongly enriched at enhancers (*Figure 2A*), with a preference for active enhancers, where it co-localizes with established enhancer marks such as H3K27ac, H3K4me1, and p300-binding (*Figure 2D*; *Creyghton et al., 2010*; *Rada-Iglesias et al., 2011*). Multiple single gene validations by ChIP-qPCR confirmed the genome-wide data and also revealed that H3K64ac levels are very low at repetitive elements (*Figure 2—figure supplement 2A,B*), where H3K64me3 is highly enriched (*Daujat et al., 2009*), again indicating opposing genomic localizations for these two marks.

To study H3K64ac in a dynamic system, we followed the kinetics of H3K64ac during ES cell differentiation (*Figure 3A*). In pluripotent ES cells, H3K64ac is strongly enriched at active pluripotency-associated genes such as *Nanog*, *Pou5f1* and *Dppa3*, whereas after retinoic acid-induced differentiation this enrichment shifts towards active differentiation-associated genes (e.g., *Hoxb3* and *Hoxd3*, *Pax6*) demonstrating that H3K64ac levels reflect transcriptional activity during differentiation. Since we found H3K64ac tightly associated with transcribed regions, we next asked if H3K64ac is enriched on

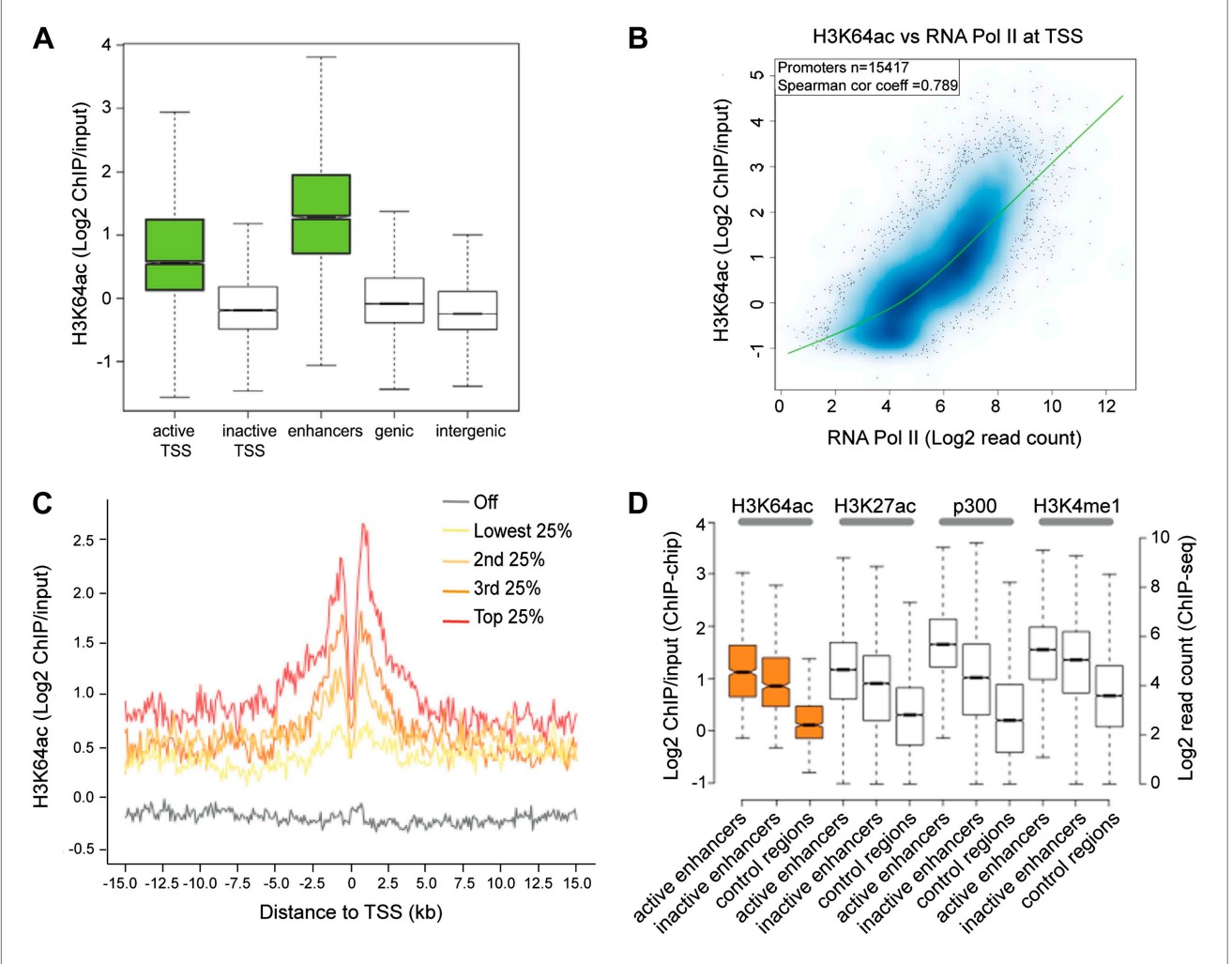

**Figure 2**. H3K64ac is enriched genome-wide at active regulatory regions. (**A**) H3K64ac is predominantly localized to active TSS and enhancers in mouse ES cells (see also **D**). Boxplot showing H3K64ac signal intensity (log2 ChIP/input) for all microarray probes at active TSS, inactive TSS, enhancers, gene bodies, and intergenic regions. (**B**) Comparison of H3K64ac and RNA Pol II at TSS. Scatterplot showing signal density distribution and global correlation. The green line is a loess-fitted trend line. (**C**) Meta-gene plot showing H3K64ac enrichment around TSS grouped according to their expression level. (**D**) Boxplot comparing H3K64ac to 'enhancer-specific' histone modifications and p300 levels (*Creyghton et al., 2010*) at active enhancers, inactive enhancers and control regions (enhancer regions shifted by 100 kb). H3K64ac levels were measured by ChIP-on-chip (left Y axis), whereas the other modifications and p300 by ChIP-seq (right Y axis).

The following figure supplements are available for figure 2:

**Figure supplement 1**. H3K64ac distribution within active chromatin.

**Figure supplement 2**. Genome-wide data validation by ChIP-qPCR.

specific H3 variants. In line with previous findings that H3.3 has covalent modifications associated with transcriptionally active chromatin (*Hake et al., 2006*), we found the highest enrichment of H3K64ac on the H3 variant H3.3 (*Figure 3B*, *Figure 3—figure supplement 1A*). Importantly these experiments also show that mutation of K64 to R results in a loss of detection of H3 by the H3K64ac antibody used (*Figure 3—figure supplement 1B*), suggesting high specificity of the antibody.

Our staining and ChIP data suggested mutually exclusive distribution patterns of H3K64 acetylation and methylation. To corroborate this in a functional model, we made use of imprinted loci; an ideal system to study potentially opposing histone marks, as one allele is transcriptionally silent whilst the other one is active (*Singh et al., 2010*). We analysed five imprinting control regions (ICRs) and in each

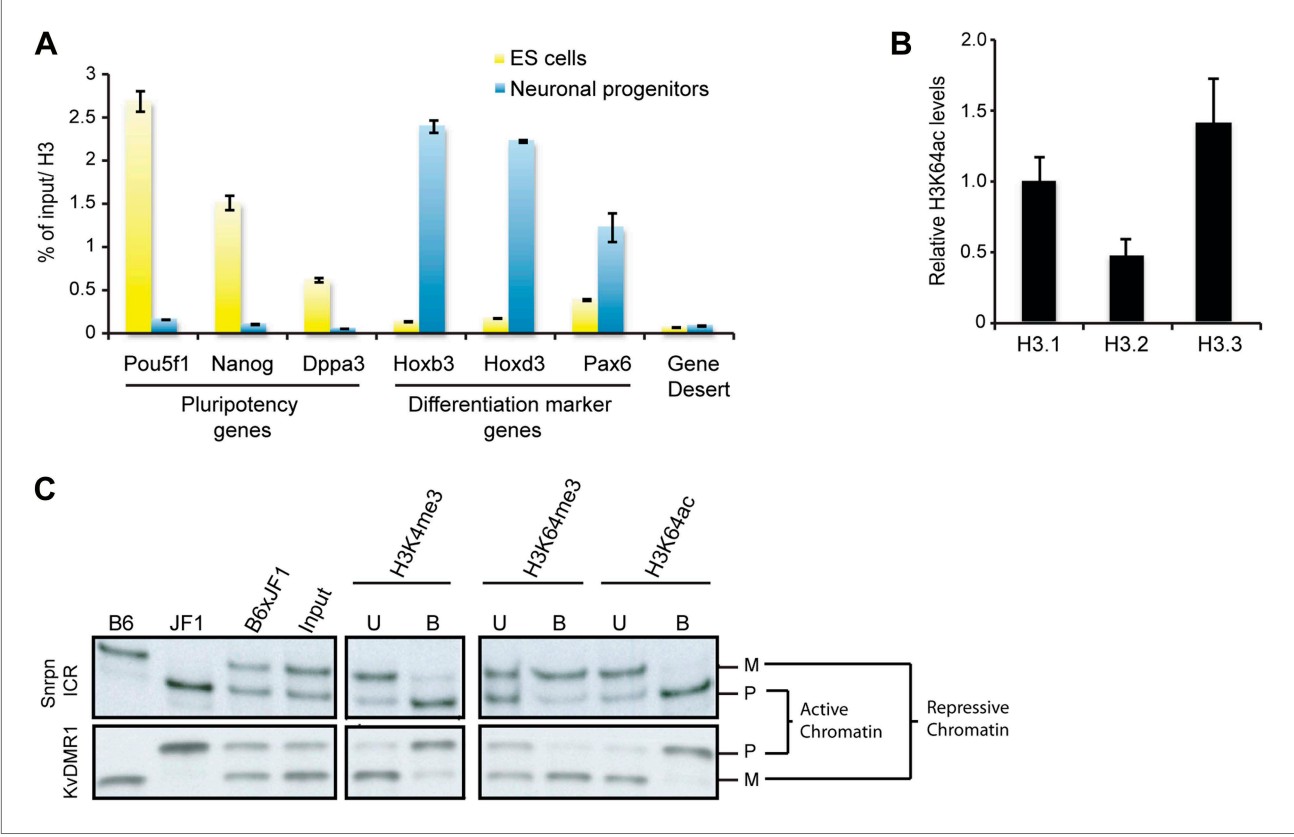

**Figure 3**. H3K64ac is enriched on active genes. (**A**) ChIP analysis of H3K64ac on pluripotency genes and differentiation-specific genes. Real-time PCR analysis for indicated promoter regions and gene desert in undifferentiated (yellow bars) or retinoic acid-induced (blue bars) ES cells. (**B**) Distribution of H3K64ac among the H3 variants. Flag-HA-tagged histones H3.1, H3.2, or H3.3 were immunoprecipitated and probed with H3K64ac antibody. Average quantification of three biological replicates (±SD) showing H3K64ac signal over HA relative to H3.1. (**C**) The active alleles of ICRs are specifically marked by H3K64ac. Native ChIP performed on primary embryonic fibroblasts of (C57Bl/6 x JF1) F1 genotype followed by radioactive PCR across polymorphic nucleotides between the paternal JF1 (*M. m. molossinus*) and the maternal C57BL/6J (B6) genomes. Single-strand conformation polymorphisms (SSCP) were revealed by electrophoresis through a non-denaturing agarose gel. The left panels show PCRs on control genomic DNAs to depict the SSCP polymorphisms used. In the input chromatin (input), the parental were equally represented at the loci analysed. U, unbound fraction; B, bound fraction; M, maternal allele; P, paternal allele.

The following figure supplements are available for figure 3:

**Figure supplement 1**. Immunoblots of H3K64ac distribution among H3 variants.

**Figure supplement 2**. The active alleles of ICRs are specifically marked by H3K64ac.

case we found that the transcriptionally active alleles were specifically enriched in H3K64ac, whereas the inactive ones were enriched in H3K64me3 (*Figure 3C*, *Figure 3—figure supplement 2*). These data suggest that H3K64ac and H3K64me3 can define functionally opposing chromatin states.

## H3K64ac can be set by p300/CBP

To identify the enzyme(s) responsible for H3K64ac, we systematically depleted candidates from different HAT families. In these assays, knockdown of p300 and CBP, but not of other HATs decreased the steady-state levels of H3K64ac (*Figure 4A*, *Figure 4—figure supplement 1A*). This decrease was most pronounced at p300/CBP-specific genomic target regions (*Figure 4B*). In line with this, overexpression of p300 resulted in increased levels of H3K64ac (*Figure 4C*, *Figure 4—figure supplement 1B*), and p300 and H3K64ac distributions showed a strong correlation (*Figure 4—figure supplement 1C*). Moreover, p300 and CBP can acetylate H3K64 in vitro on free H3 (*Figure 4D*) and within chromatin (*Figure 4—figure supplement 1D*). Altogether these data clearly establish p300/CBP as H3K64 acetyltransferases, not excluding the presence of additional H3K64 acetyltransferases.

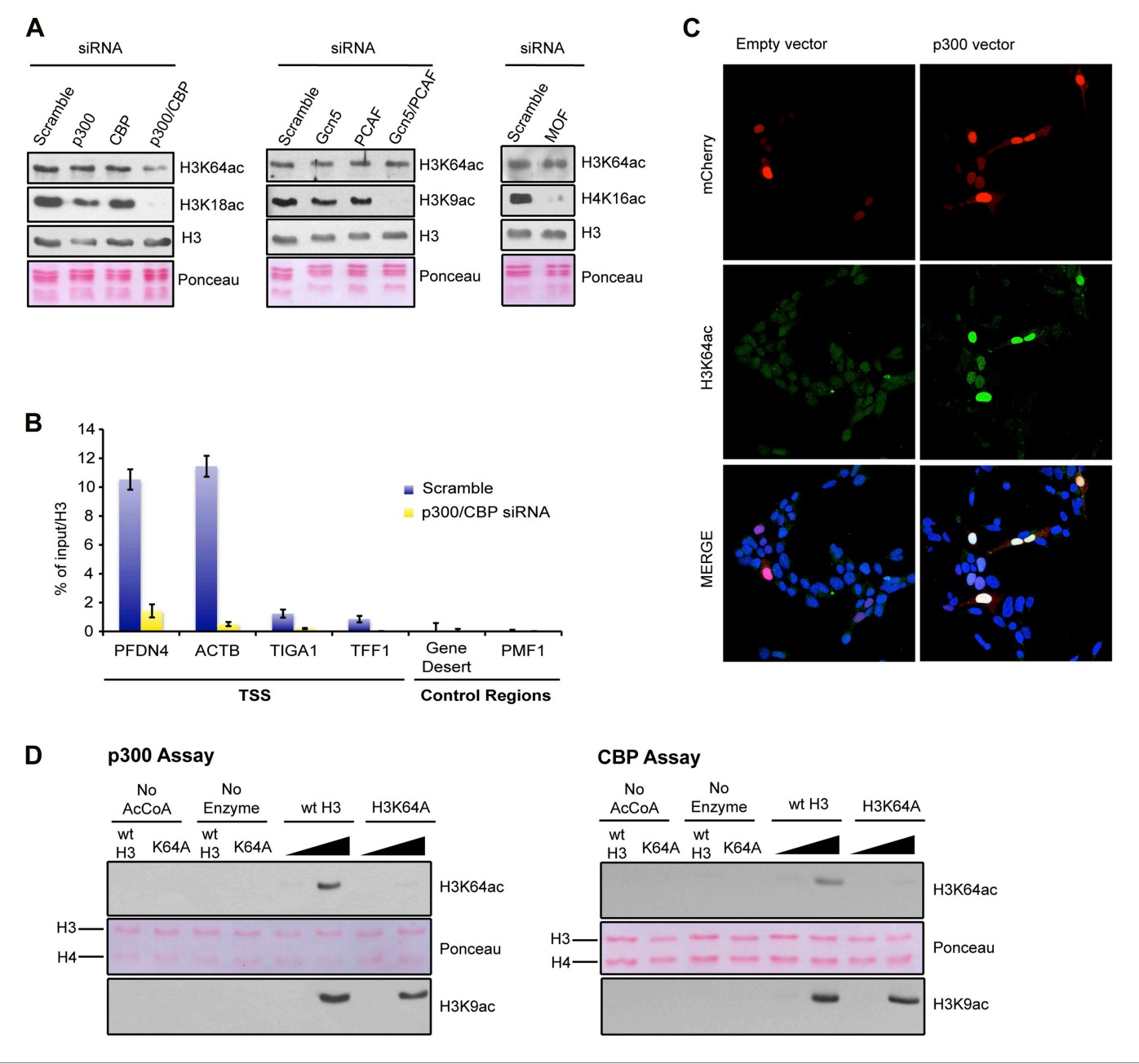

**Figure 4**. p300 acetylates H3K64 in vivo and in vitro. (**A**) siRNA-mediated depletion of HATs (as indicated) in MCF7 cells. Immunoblot analysis of global H3K64ac levels and additional modifications as controls for siRNA efficiency. Anti-H3 blot and Ponceau staining are shown as loading controls. (**B**) ChIP analysis of H3K64ac enrichment on different mouse genomic regions (as indicated) upon depletion of p300/CBP (yellow) compared to the control knock-down (scramble, blue). (**C**) Overexpression of p300 in HEK293 cells. Control (empty vector) or p300 overexpressing cells co-expressed mCherry (red, top panel) and were assessed for H3K64ac levels (green) in immunoflourescence. (**D**) In vitro HAT assay with p300 or CBP using recombinant H3 (wt or K64A mutant) as substrate and probed with the H3K64ac antibody (top panel). Ponceau staining as loading control (middle panel) and H3K9ac western blot as activity control (bottom panel) are shown.

The following figure supplements are available for figure 4:

**Figure supplement 1**. Validation and supporting experiment to establish p300 activity on H3K64.

## H3K64ac can increase nucleosome instability

Given the location of H3K64ac on the nucleosome's lateral surface, a potential mechanism is that it acts through modulating ATP-dependent chromatin remodelling and/or nucleosome stability. To investigate this, we produced recombinant histone H3 acetylated on K64 in *Escherichia coli* using site-specific genetically directed incorporation of acetyl-lysine (*Neumann et al., 2009*; *Figure 5—figure supplement 1A,B*). To address effects on chromatin remodelling, we incubated H3K64ac and unmodified nucleosomes with two different chromatin remodelers belonging to two different Snf2 subfamilies, Chd1 and RSC (*Flaus et al., 2006*; *Clapier and Cairns, 2009*). Our results show that Chd1 repositioned H3K64-acetylated nucleosomes faster than unmodified nucleosomes. This was not the case when we used the RSC enzyme (*Figure 5—figure supplement 2*). This suggests that acetylation of H3K64 could differentially affect remodelling enzymes. Next, we sought to investigate whether H3K64ac also impacts on passive fluctuations in nucleosome structure and interrogated whether H3K64ac affects the stability of DNA association with histone octamers within nucleosomes. To do this, we attached fluorescent dyes to specific sites on the DNA, 35 bp from each end of the nucleosomal DNA, and performed FRET measurements (*Neumann et al., 2009*). Using this strategy, we observed that the FRET interaction for H3K64ac nucleosomes was more sensitive to salt-disruption than unmodified nucleosomes at NaCl concentrations ranging between ~0.5 M and 1.0 M (*Figure 5A*). In parallel, we also measured the salt-dependent nucleosome stability by single-molecule FRET, which again resulted in a lower stability of H3K64ac nucleosomes (*Figure 5—figure supplement 3*). Together these data demonstrate a decreased stability of H3K64ac nucleosomes, distinguishing this acetylation from H3K56ac that was reported not to significantly affect nucleosome stability under comparable conditions (*Neumann et al., 2009*).

Modifications of histones that affect interactions between the H3–H4 tetramer and DNA also have the potential to influence the efficiency of chromatin assembly. To address whether H3K64ac affects the affinity of histone–DNA binding, we performed competitive nucleosome reconstitution assays (*Thåström et al., 2004*). *Figure 5B* shows that H3K64ac reduces the level of nucleosome assembly, relative to unmodified H3, indicating that H3K64 acetylation reduces histone–DNA binding affinity. In contrast, acetylation at H3K9 had no detectable effect (*Figure 5A,B*), illustrating functional differences between acetylation of lateral surface and histone tail lysines. In concert, these observations demonstrate that H3K64ac directly influences histone–DNA association and nucleosome stability. This could provide a mechanistic explanation as to why H3K64ac is found in vivo at sites where increased nucleosome mobility/instability is required, such as the TSS of active genes (*Segal et al., 2006*).

In vivo, one of the most dramatic chromatin-reorganization events that require nucleosome instability occurs during spermatogenesis. In mammalian elongating spermatids, histones are initially replaced by the transition proteins and then by protamines (*Gaucher et al., 2010*, *2012*). In spermatocytes and round spermatids H3K64ac was below the limit of detection. However, the levels of H3K64ac dramatically increased in elongating spermatids (*Figure 5C,D*). At approximately the same time, we also detected the presence of some specific H3 tail acetylations (*Figure 5—figure supplement 4A*). Notably, we observed this increase in H3K64ac precisely during the period of nucleosome disassembly and replacement of histones by transition proteins in spermiogenesis. Our present data suggest a potential role for lateral surface modifications, such as H3K64ac, in this replacement process by creating a less stable, more 'open' chromatin state. In contrast to this, H3K64me3 is enriched in the part of the spermatid where genomic regions that undergo late histone replacement are located, suggesting that H3K64me3 may 'stabilize' particular regions of chromatin thereby protecting them from histone exchange (*Figure 5—figure supplement 4B*).

## H3K64ac can promote histone eviction and facilitate transcription

To test whether H3K64ac can indeed promote histone displacement from chromatin, we assembled chromatin with unmodified H3, H3K64R, or H3K64ac octamers on a template containing five GAL4-binding sites in front of a MLP promoter (*Figure 6—figure supplement 1*) and performed a histone eviction assay (*Figure 6A*, left panel). Using this assay, we detected more H3 in the evicted fraction from H3K64ac chromatin than from H3K64R chromatin (*Figure 6A*, right panel, *Figure 6—figure supplement 2A*), suggesting that the destabilization introduced by H3K64ac facilitates histone displacement and eviction. This could explain at least in part how H3K64ac can directly impact on nucleosome stability at active promoters and is in line with our observation that H3K64ac is associated with regions of the genome that are transcriptionally active or require histone exchange/turnover. Of note, we detected equivalent binding of Nap1 to each of the H3 species used (unmodified, K64R, or K64ac) indicating that the observed difference in histone eviction is not due to differential Nap1 histone binding (*Figure 6—figure supplement 2B*).

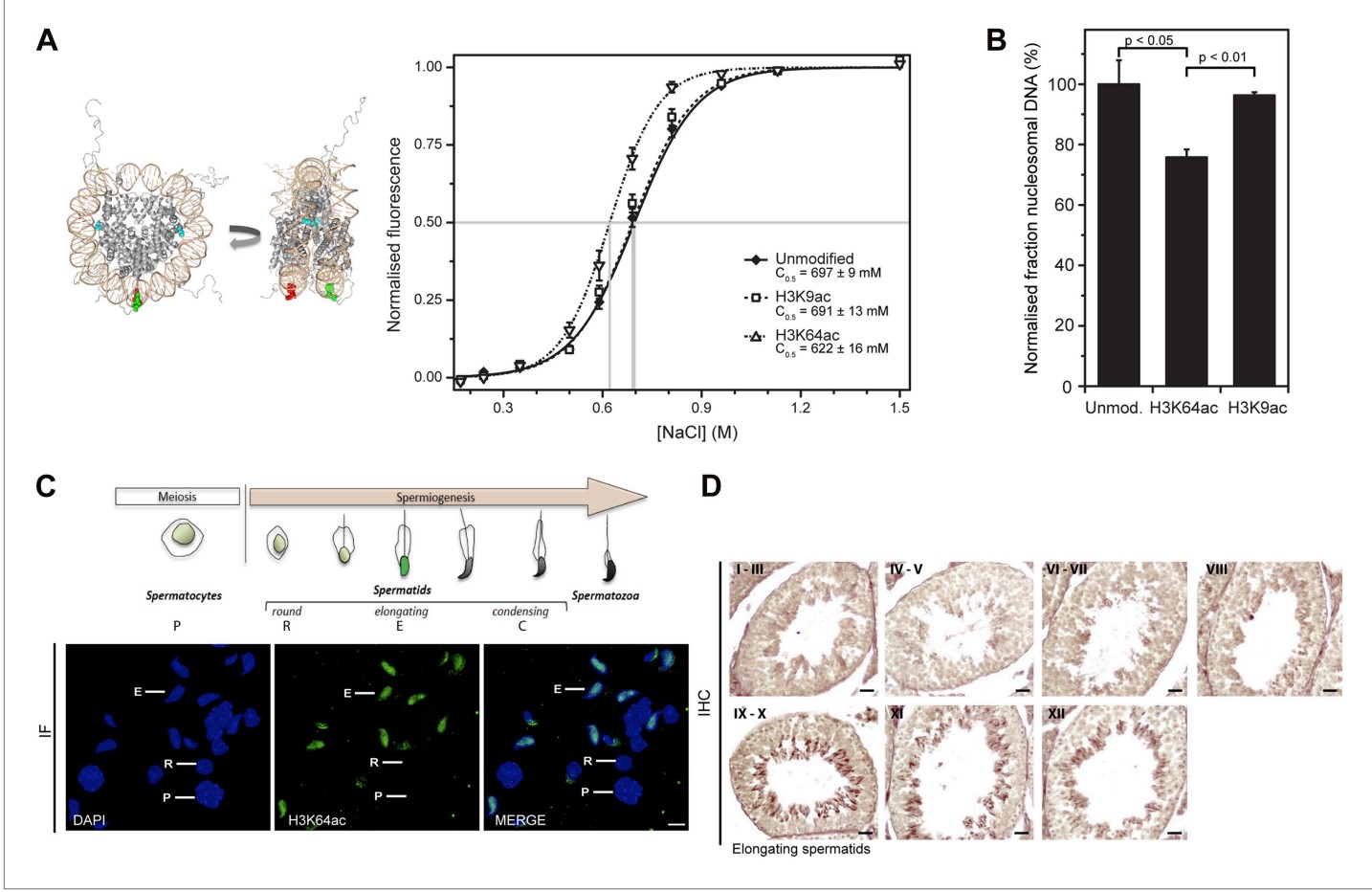

**Figure 5**. H3K64 acetylation affects DNA-octamer interactions. (**A**) H3K64ac nucleosomes are more sensitive to salt-induced disruption in in vitro FRET assays. Left, different views of a nucleosome with H3K64 highlighted in blue (spacefill) and the internal positions of the dye-labeled bases 35 bp from the end of the nucleosome. Right, the average (±s.e.) of three separate titrations is shown for unmodified (black squares), H3K9ac (unfilled squares), and H3K64ac nucleosomes (unfilled triangles). (**B**) Competitive in vitro nucleosome assembly reactions suggest that H3K64ac weakens histone–DNA interactions. Unmodified, H3K9ac, and H3K64ac nucleosomes were assembled in the presence of an excess of non-specific competitor DNA. The average (±s.e.) of three independent competitive assembly reactions is shown. (**C**) In spermatogenesis H3K64ac levels are high in elongating spermatids during the wave of massive chromatin remodelling. Schematic of mouse spermatogenesis (top). Immunofluorescence stainings (IF) show high H3K64ac levels in elongating spermatids (E) and below detection limit in spermatocytes (P) and round spermatids (R). Scale bar, 5 μm. (**D**) Immunohistochemistry stainings (IHC) on sections of mouse testis tubules. H3K64ac (brown) is enriched in elongating spermatids (stages IX-X), which are present near the lumen. Scale bar, 20 μm.

The following figure supplements are available for figure 5:

**Figure supplement 1**. Validation of the substrates used in the in vitro assays.

**Figure supplement 2**. Effect of H3K64ac on repositioning by RSC and Chd1.

**Figure supplement 3**. Single-molecule analysis of unmodified and H3K64ac salt-dependent nucleosome stability.

**Figure supplement 4**. H3 acetylation and H3K64me3 stainings in spermatogenesis.

**Figure supplement 5**. Structural analysis of H3K64 interactions.

Finally, to demonstrate a function for H3K64ac in gene expression in vivo, we expressed wildtype H3.3, H3.3K64Q (acetylated lysine mimetic) and H3.3K64R (retains positive charge but is non-acetylatable) in NIH3T3 cells (*Figure 6—figure supplement 3A*) and assayed for effects in TPA-dependent early-response gene activation. In this setup, H3K64Q promoted the expression of certain genes, such as *c-fos*, *Egr1*,

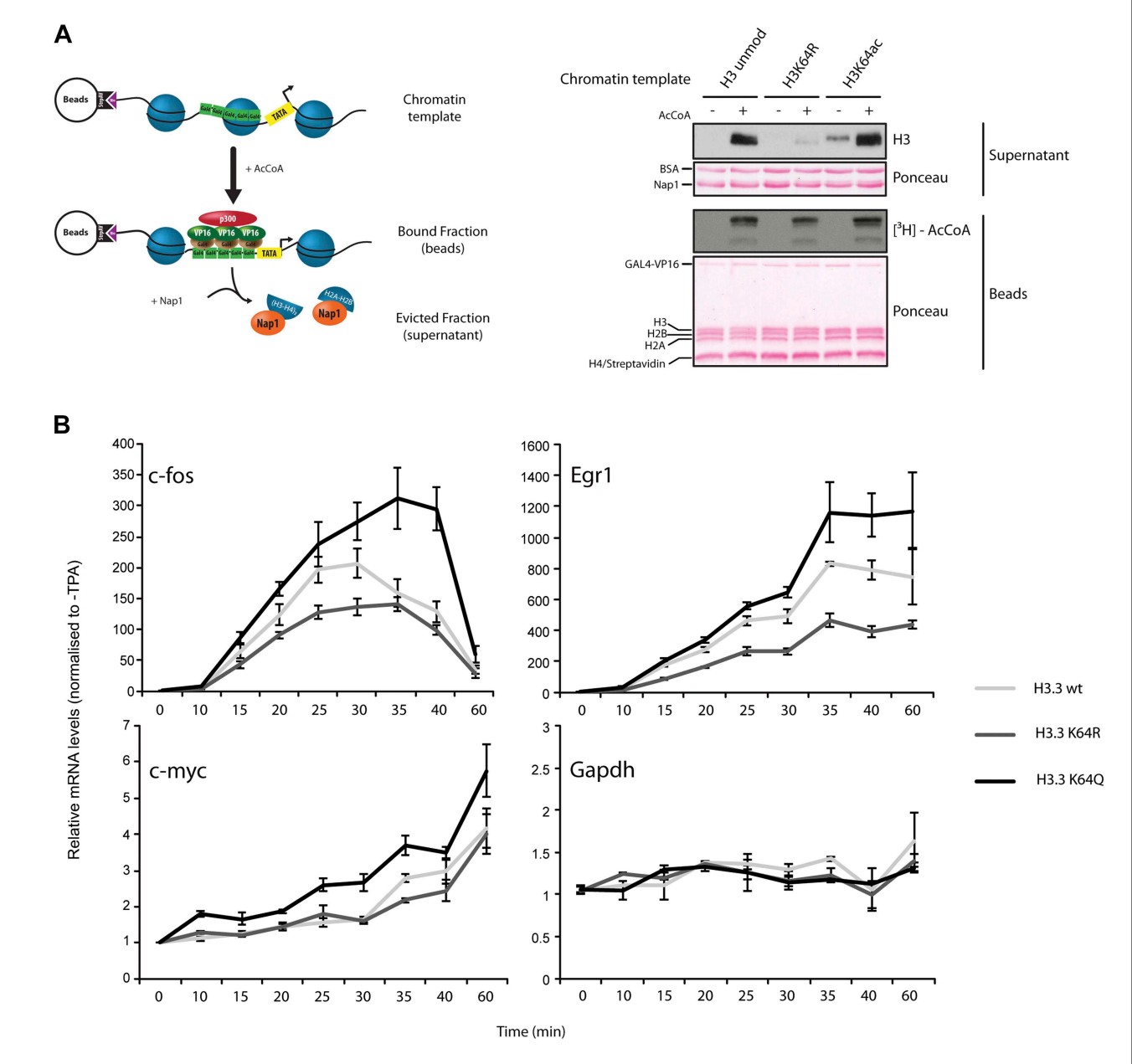

**Figure 6**. H3K64ac can promote histone eviction and facilitate transcription. (**A**) H3K64ac promotes histone eviction. Schematic of eviction assay (left) shows chromatin templates assembled on a biotin-labeled pG5-MLP promoter fragment. Recruitment of transcriptional activators and p300 causes displacement of histone octamers, and histones can be trapped onto a supercoiled plasmid (not shown) by Nap1. Immunoblot of the evicted fraction (right, supernatant) with anti-H3 antibody shows the total amount of H3 released from unmodified H3, H3K64R, or H3K64ac chromatin templates in the presence or absence of acetyl-CoA. Ponceau staining as loading control and autoradiography ([3H]-acetyl-CoA incorporation) on the immobilized fraction as control for p300 activity are shown. (**B**) The acetylation mimic H3K64Q promotes transcription of early-response genes. NIH3T3 cells overexpressing either wild type, K64R, or K64Q FlagHA-H3.3 were stimulated with 25 nM TPA for the indicated times. mRNA levels of the indicated genes were analysed by qPCR. Gapdh is shown as control gene. Average of three experiments (±s.e.) are shown.

The following figure supplements are available for figure 6:

**Figure supplement 1**. Validation of the chromatin templates used in the in vitro assay.

**Figure supplement 2**. Control experiments for the eviction assay involving the role of Nap1.

**Figure supplement 3**. Controls for H3.3 expression and gene induction.

and *c-myc*, (*Figure 6B*), above the levels obtained when assaying wildtype H3.3 or the acetylation-deficient H3.3K64R. Notably, expression of control genes such as *Gapdh* and *Hsp70* was unaffected (*Figure 6B*, *Figure 6—figure supplement 3B*). The observed transcriptional effects due to a single acetylation mimic within the globular domain strongly suggest that H3K64ac has an intrinsic ability to impact upon transcription mechanisms in vivo. Taken together, our results provide novel mechanistic insights into how a single histone modification, H3K64ac, within a nucleosome could specifically regulate nucleosome stability and transcription.

## Discussion

We have identified and functionally characterized a novel histone modification located on the octamer's lateral surface in close proximity to the DNA backbone of the inner gyre. The main chain atoms of H3K64 are in direct contact with the DNA phosphate backbone and this places the H3K64 side chain close enough to potentially contact the DNA, with the average H3K64 side-chain nitrogen-to-DNA distance approximately 6.1 Å (*Figure 5—figure supplement 5A,B*). Lysine acetylation results in increased bulk and charge neutralization of the lysine side-chain, and it is likely that acetylation of lateral surface lysines (e.g., H3K64) could interfere with histone–DNA interactions. Interestingly, in the high-resolution crystal structure of the nucleosome (PDB 1KX5; *Davey et al., 2002*) H3K64 is found to be involved in an extensive network of water-mediated hydrogen bonds that link numerous water molecules, histone side- and main-chain atoms and the DNA (*Figure 5—figure supplement 5C*). Acetylation of H3K64 would be predicted to disrupt this hydrogen bond network. Davey et al. estimate from their crystallographic analysis that water-bridged interactions between histones and DNA are as important to nucleosome stability as the direct histone–DNA contacts (*Davey et al., 2002*). These water-bridges provide a plasticity that allows the histone core to adapt to myriad variations in the DNA sequence and conformation. They postulate that disruption of even a single water–bridge interaction (e.g., acetylation of H3K64) has the potential to significantly affect nucleosome stability. This is indeed fully supported by our findings—we demonstrate that H3K64ac exerts a direct effect on nucleosome stability and facilitates histone eviction.

H3K64 acetylation could cause the reduced FRET signal we observed in the salt-dependent dissociation assay in multiple ways. A pre-requisite for the loss of histones from the nucleosome is the dissociation of DNA from the surface of the histone octamer. Given the proximity of H3K64ac to DNA it is likely that it may facilitate this dissociation. This loss of histone–DNA contacts caused by H3K64ac may aid the loss of H2A–H2B dimers, contributing to the observed change in FRET signal. Alternatively, following the loss of H2A–H2B dimers H3K64ac may favour dissociation of DNA from histones H3 and H4, or H3K64ac may affect water-mediated hydrogen bonds that directly influence the association of H2A–H2B dimers. These modes of action are not mutually exclusive and they are consistent with the observed effects of H3K64ac in chromatin assembly in vitro. They also all have the potential to influence chromatin organization during cycles of assembly/disassembly at gene regulatory elements. We detect an approximately 60 mM decrease in the salt-dependent stability between H3K64ac and unmodified/H3K9ac nucleosomes. This difference in salt stability of H3K64ac appears to be in a range consistent with the observed changes in nucleosome dynamics. *Gansen et al. (2013)* showed that simultaneous acetylation of all sites in H3 results in a decrease in nucleosome salt stability of approximately 100–130 mM NaCl. Furthermore, a fundamental change in the histone content of the nucleosome via incorporation of the histone H2A variant H2A.Z, increases salt stability by up to ~70 mM (*Park et al., 2004*). Moreover, the observed enrichment of H3K64ac at active gene promoters and during chromatin reorganization in spermiogenesis argues for a role of H3K64ac in creating an open, permissive chromatin state. Indeed, this is supported by increased histone eviction and the finding that a mutation mimicking constitutive acetylation, H3K64 to Q, can promote transcription of at least certain genes. We consider a twofold higher transcription due to a single histone mutation as biologically important.

Additional lysines on the octamer's lateral surface have been found to be acetylated, such as H3K56, which is located closer to the ends of the nucleosomal DNA. Interestingly, these acetylations can mediate distinct effects on nucleosomes (this work and *Neumann et al., 2009*; *Tropberger et al., 2013*). Thus, specificity exists in the way that the biomechanical properties of nucleosomes are affected differentially by acetylation at different sites within the globular domain of H3.

Despite considerable efforts, we have so far been unable to detect specific readers, or binders of H3K64ac. Consequently, we prefer a model in which it acts via direct effects. However, H3K64ac may also function by preventing H3K64me3 from being laid down at key genomic regions. We have previously suggested that methylation of H3K64 is involved in the creation of repressive chromatin states (*Daujat*

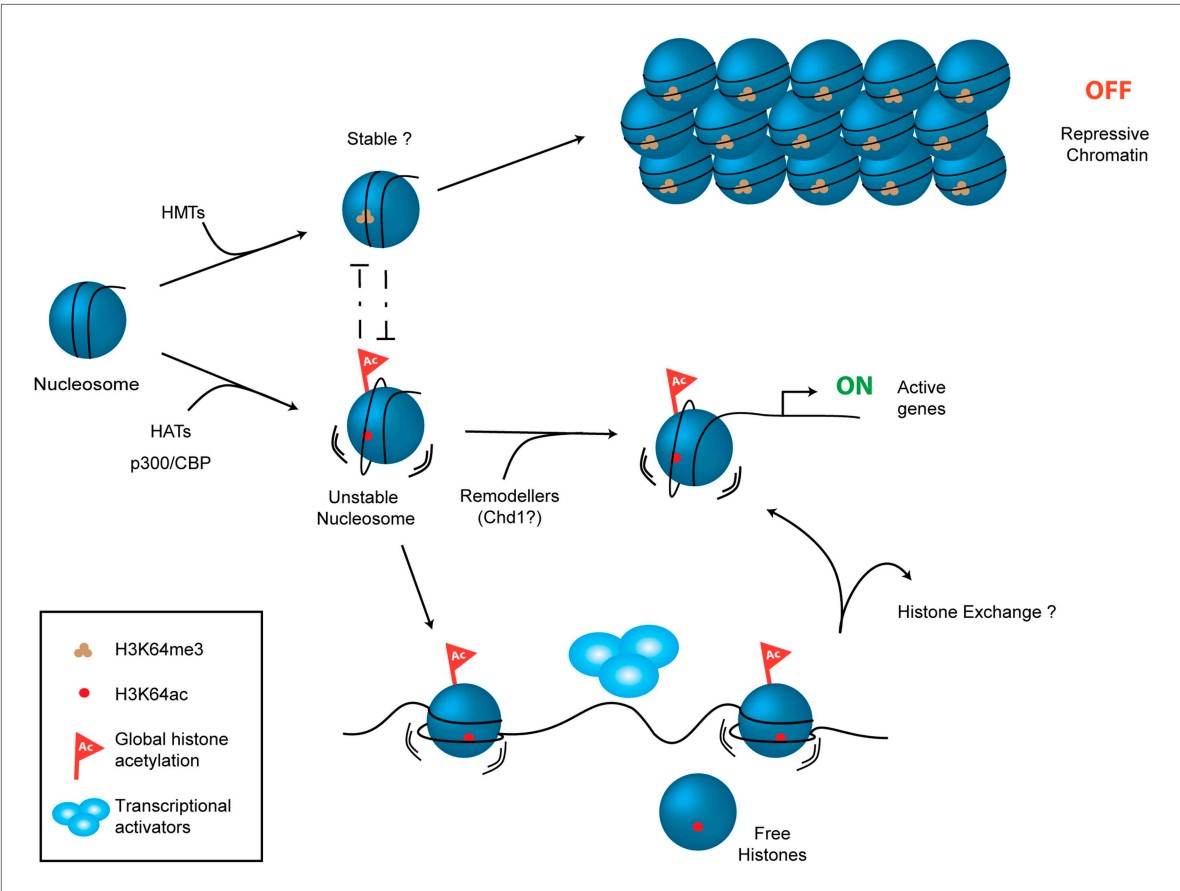

**Figure 7**. Schematic model for H3K64ac function. H3K64ac (bottom) helps to create an open, permissive chromatin environment. In contrast to this H3K64me3 could lock the nucleosomes in a stable, repressive conformation (top), resulting in an inactive chromatin environment. H3K64ac can directly act by affecting nucleosome dynamics, but also indirectly by blocking methylation of the same residue.

*et al., 2009*; *Lange et al., 2013*). Thus, H3K64ac might serve to prevent the nucleosome from adopting a repressive conformation, thereby maintaining chromatin in a transcriptionally competent state. Our working model (*Figure 7*) proposes that opposing chromatin states can be defined via the mutually exclusive presence of H3K64ac or H3K64me3. This model is reinforced by the observed distinct and largely mutually exclusive genomic distribution patterns of these two H3K64 modifications.

The data provided here strengthen the concept that lateral surface modifications play a key role in regulating chromatin function, as initially proposed by others (*Cosgrove et al., 2004*; *Mersfelder and Parthun, 2006*) and us (*Tropberger and Schneider, 2013*). They also extend our understanding of chromatin regulation and provide mechanistic insights into how chromatin organization is spatially and temporally controlled by globular domain modifications. The identification of novel pathways regulating chromatin function has also the potential to highlight druggable targets for diseases in which chromatin states have been distorted.

## Materials and methods

### MS/MS analysis of acetylated H3

In vivo modified histone H3 was digested either with trypsin or chymotrypsin and analyzed by nanoLC-MS as previously published (*Waldmann et al., 2011*). For both digests, STAGE tip-assisted sample purification was achieved essentially as described (*Rappsilber et al., 2007*). Desalted samples were subsequently analyzed using nanoflow (Agilent 1200 nanoLC, Germany) LC-MS/MS on a linear ion trap (LIT)-Orbitrap (LTQ-Orbitrap XL) mass spectrometer (ThermoFisher, Germany). Peptides were eluted with a linear gradient of 10–60% buffer B (80% ACN and 0.5% acetic acid) at a flow rate of 250 nl/min

over 40 or 60 min depending on the experiment. Data were acquired using a data-dependent 'top 5' method, dynamically choosing the five most abundant precursor ions from the survey scan (mass range 250–1650 Th) in order to isolate and fragment them in the LTQ. All data were acquired in the profile mode and dynamic exclusion was defined by a list size of 500 features and exclusion duration of 30 s with a MMD of 10 ppm. Early expiration was disabled to decrease the resequencing of isotope clusters. The isolation window for the precursor ion selection was set to 2.0 Th. Precursor ion charge state screening was enabled, and all unassigned charge states as well as singly charged ions were rejected. For the survey scan a target value of 10,00,000 (1000 ms maximal injection time) and a resolution of 60,000 at $m/z$ 400 were set (with lock mass option enabled for the 445.120024 ion), whereas the target value for the fragment ion spectra was limited to 10,000 ions (200 ms maximal injection time). The general mass spectrometric conditions were: spray voltage, 2.3 kV; no sheath and auxiliary gas flow; ion transfer tube temperature, 150°C; collision gas pressure, 1.3 mTorr; normalized collision energy using wide-band activation mode; 35% for MS$^2$. Ion selection thresholds were 500 or 1000 counts for MS$^2$ depending on the experiment. An activation q = 0.25 and activation time of 30 ms was applied.

MS data were processed into peak lists by DTASuperCharge 2.0b1 (part of the MSQuant 2.0b7 software environment *Mortensen et al., 2010*) and searched with Mascot 2.2 against the human International Protein Index protein database (IPI, version 3.65) combined with frequently observed contaminants and concatenated with the reversed versions of all sequences. The MMD for monoisotopic precursor ions and MS/MS peaks were restricted to 5 ppm and 0.8 Da, respectively. Enzyme specificity was set to trypsin (with a maximum of two missed cleavages) allowing cleavage N-terminal to proline and C-terminal to aspartate. Modifications were cysteine carbamidomethylation (fixed) and protein lysine acetylation, lysine butyrylation, lysine propionylation, lysine and arginine methylation (all states), lysine formylation, serine/threonine/tyrosine phosphorylation, deamidation (asparagine and glutamine) and methionine oxidation (variable). Alternatively, we searched the chymotrypsin digested sample with low chymotrypsin specificity (cleavage carboxyterminal to L, M, W, F, and Y allowing a maximum of four missed cleavages). Protein and peptide identifications were further analyzed and manually verified by inspection of chromatograms and spectra.

To confirm the identity of the tryptic peptide K(Ac)LPFQR (*Figure 1A*) a synthetic peptide (Biosyntan, Berlin) harbouring the sequence STELLIRK(ac)LPFQRLVGC was trypsin digested and demonstrated the generation of three tryptic peptides, in particular the peptide K(ac)LPFQR. The MS/MS fragment spectrum of the latter was virtually identical to the one derived from the endogenous tryptic peptide. Both spectra contain two significant immonium ion peaks (at m/z 143 and m/z 126) that indicate the presence of acetylated lysine at the amino terminus (*Trelle and Jensen, 2008*). In addition two chymotryptic peptides with the sequence LIRK(Ac)LPF (*Figure 1A*) and IRK(Ac)LPF (*Figure 1—figure supplement 1A*) confirm the presence of an acetylated lysine at K64 of endogenous histone H3.

## 3D modeling

The acetylation of H3K64 was modelled on the 1KX5 *Xenopus laevis* nucleosome core particle structure (*Davey and Richmond, 2002*). Pictures and modifications were generated using PyMOL (http://www.pymol.org).

## Cell culture

Mouse embryonic stem (ES) cells used for chromatin immunoprecipitation (ChIP) experiments were derived and cultivated as described previously (*Mohn et al., 2008*). ES cells differentiation was carried out as described previously (*Bibel et al., 2007*). Briefly, embryonic stem cells were cultured feeder-free for three to five passages after which LIF was withdrawn to allow formation of cellular aggregates in liquid culture. After 4 days, retinoic acid (RA) was added to induce neuronal progenitor (NP) formation for another 4 days before chromatin isolation. All the other mammalian cell lines were cultured in DMEM (PAA, GE Healthcare Life Sciences, Sweden) medium supplemented with 10% FCS, 2 mM L-glutamine, 100 U/ml Penicillin and 0.1 mg/ml Streptomycin. For the HDAC inhibitors treatment either Na-butyrate (10 mM, overnight) or Trichostatin A (TSA, 0.2 µM, 48 hr) were used.

## Antibody characterization

For characterization of the H3K64ac antibody (Active motif, Carlsbad, CA, cat no. 39545, lot 32908001) immuno-dot blotting and peptide competitions were carried out as described previously (*Daujat*

*et al., 2009*). Histone H3 acetylated peptides corresponding to amino-acid residues 1–15 (for K9), 12–24 (for K18), 1–27 (for K23), 1–27 (for K9-K14-K23), and 51–62 (for K56) and histone H3 methylated peptides corresponding to amino-acid residues 57–71 (for K64me1, me2 and me3) were used in these assays.

## Mononucleosome preparation and limited trypsin digestion

$1 \times 10^8$ mouse NIH3T3 cells were harvested, washed in PBS, resuspended in cold hypotonic buffer (5 mM KCl, 1.5 mM $MgCl_2$, 20 mM Hepes pH 7.5, 1x protease inhibitors (Roche, Switzerland), 5 mM Na-butyrate) at a density of $2 \times 10^7$ cells/ml and pelleted. Cells were resuspended in the same buffer at $4 \times 10^7$ cells/ml and incubated on ice for 10 min. Cell suspension was dounced 12 times with the S pestle and store on ice for 30 min. After 10 min of centrifugation at $500 \times g$ at 4°C, released nuclei were resuspended in 1 ml of isolation buffer-100 (0.25 M sucrose, 100 mM NaCl 1.5 mM $MgCl_2$, 1 mM $CaCl_2$, 10 mM Tris–HCl pH 7.5, 1x protease inhibitors, 5 mM Na-butyrate) and stored 10 min on ice. After 5 min of centrifugation at 4°C nuclei were resuspended in 1 ml of isolation buffer-250 (same as previously with 250 mM NaCl) and stored 10 min on ice. Nuclei were centrifuged at $4000 \times g$ for 10 min at 4°C and resuspended in 1 ml of isolation buffer-0 (same as previously with no NaCl) for DNA concentration determination. The chromatin was then digested in the same buffer for 30 min at 37°C (50 units/mg DNA of Micrococcal Nuclease in the presence of 3 mM $CaCl_2$) and stopped by the addition of 10 mM EDTA. Nuclei were centrifuged at $10000 \times g$ for 10 min and resuspended in 1 ml of lysis buffer (650 mM NaCl, 5 mM EDTA pH 8.0, 10 mM Tris–HCl pH 6.8, 1x protease inhibitors, 5 mM Na-butyrate) and incubated on ice for 30 min. Lysed nuclei were centrifuged at $16500 \times g$ for 10 min at 4°C and supernatant containing digested chromatin was loaded on a 5–40% sucrose gradient and centrifuged at 40,000 rpm (SW41 rotor) for 16 hr at 4°C. 600 µl fractions were then collected and the concentration of chromatin was determined at OD 260 nm. Chromatin of each fraction was either analysed by electrophoresis on a 1% agarose gel in 0.5x TBE or by SDS-PAGE to assess DNA size and core histones integrity. Fractions containing mononucleosomes were pooled and used for limited trypsin digestion.

10 µg of DNA corresponding to mononucleosomes were digested by 20 and 30 µg of trypsin for 30 min at 26°C in 250 µl final of PBS. This mild digestion cuts the N-terminal histones tails protruding out from the nucleosome and leaves the H3 core region (protected by the DNA) mainly intact. Intact or digested nucleosomes were analysed by SDS-PAGE and Western blotting for the presence of N-terminal H3 acetylations (anti-H3K9ac; anti-H3K18ac; anti-H3K27ac) and H3K64 acetylation. H3 digestion pattern was controlled with an anti-H3 antibody.

## ChIP analysis

ChIPs of histone modifications on native or crosslinked chromatin were performed with minor modifications of procedures described previously (*Cuthbert et al., 2004*; *Daujat et al., 2009*).

For native ChIPs nuclei were extracted and isolated over a sucrose cushion, resuspended in MNase buffer (0.32 M sucrose, 50 mM Tris-HCl pH 7.5, 4 mM $MgCl_2$, 1 mM $CaCl_2$) and digested with 10 U MNase (Thermo Fisher Scientific Inc., Waltham, MA) for at least 15 min at 37°C, so that mononuceosomes were released. This chromatin was incubated overnight with specific antibodies and then precipitated with protein A/G Sepharose 4 fast flow beads (GE Healthcare Life Sciences). After washing three times with increasing amount of salt (50 mM Tris-HCl pH 7.5, 10 mM EDTA, containing either 75 mM, or 125 mM, or 175 mM NaCl), immunoprecipitated DNA was eluted and purified.

For ChIPs on cross-linked chromatin ES cells were lysed and sonicated to produce chromatin fragments of 0.5 Kb on average. Diluted chromatin equivalent to $2 \times 10^6$ cells was subjected to immunoprecipitation overnight. Sepharose beads were used to recover chromatin. Precipitates were washed 10 min at RT two times in lysis buffer (50 mM HEPES/KOH pH 7.5, 500 mM NaCl, 1 mM EDTA, 1% Triton X-100, 0.1% DOC, 0.1% SDS), once in DOC buffer (10 mM Tris pH 8, 0.25 mM LiCl, 0.5% NP40, 0.5% DOC, 1 mM EDTA) and once in TE and DNA was recovered as described previously (*Dedon et al., 1991*; *Orlando et al., 1997*; *Daujat et al., 2002*). Quantitative real-time PCR were performed using SybrGreen (Thermo Fisher Scientific Inc.) on an ABI Prism 7300 apparatus (Applied Biosystems, France). The 5′ to 3′ sequences of the ChIP primers used for different genomic regions in qPCR are listed in *Supplementary file 1*.

ChIP-on-chip of histone modifications were carried out as described previously (*Mohn et al., 2008*).

For ChIP followed by PCR-SSCP to distinguish the parental alleles, primary MEFs were derived from 13.5 d.p.c. (C57BL/6 x JF1)F1 foetuses (genotype: [C57BL/6 x *M. m. molossinus*]F1). After ChIP, DNA was extracted from the immunoprecipitated (B, bound) and unbound (U) chromatin. As a control mock precipitation, we used a rabbit IgG antiserum against chicken IgI (Sigma-Aldrich, St. Louis, MO, cat no. C2288, lot 21K4851). The maternal (M) and paternal (P) alleles were distinguished by radio-active PCR amplification across nucleotides that were polymorphic between the paternal JF1 (*M. m. molossinus*) and the maternal C57BL/6J (B6) genome. Primers used for PCR are listed in *Supplementary file 1*.

Following denaturation of the amplification products, single-strand conformation polymorphisms (SSCP) were revealed by electrophoresis through a non-denaturing agarose gel (*Gregory and Feil, 1999*; *Umlauf et al., 2004*; *Pannetier et al., 2008*).

## Microarray design, hybridization and analysis

ChIP samples were hybridized to custom tiling microarrays (*Lienert et al., 2011*) representing all well-annotated promoters and the complete chromosome 19 with an average probe spacing of 100 bp (NimbleGen Systems Inc., Madison, WI).

Sample labelling, hybridization and array scanning were performed according to the manufacturer's protocols using a MAUI hybridization station and a NimbleGen MS200 slide scanner in combination with NimbleScan software (NimbleGen Systems Inc.). Further processing was done using R. For analysis raw fluorescent intensity values were used to calculate log2 of the bound/input ratios for each individual oligo. Subsequently, for comparison all arrays were normalized to a median log2 = 0 and scaled to have the same median absolute deviation using the 'LIMMA' R/Bioconductor package (*Smyth and Speed, 2003*; *Smyth, 2004*). For boxplots, all microarray probes that overlap the indicated genomic regions were grouped and log2 ChIP/Input values for every probe are displayed as one boxplot per class. Active and inactive TSS (activity cutoff defined according to *Lienert et al., 2011*) contain probes that are in a 2-kb window (± 1 kb) around the respective transcription start sites, 'enhancers' contains all probes overlapping the enhancer regions as defined by *Creyghton et al. (2010)*, 'genic' represents all probes that map to annotated genes without the first 1000 bp to avoid overlap with the promoters, and 'intergenic' contains all remaining probes on chromosome 19 that are not in any of the other groups to avoid overlap between the individual classes.

The data discussed in this publication have been deposited in NCBI's Gene Expression Omnibus (*Edgar et al., 2002*) and are accessible through GEO Series accession number GSE35355 (http://www.ncbi.nlm.nih.gov/geo/query/acc.cgi?acc=GSE35355).

## Deep sequencing data and analysis

H3K27ac and p300 ChIPseq data were taken from *Creyghton et al. (2010)*, H3K4me1 data from *Stadler et al. (2011)*, mRNA expression data from *Lienert et al. (2011)*. Raw data was downloaded from the NCBI Gene Expression Omnibus and the reads were mapped to the *Mus musculus* genome (mm9) and transcriptome (RefSeq, downloaded on 07/17/2009) as previously described (*Lienert et al., 2011*).

For promoter analysis, mapped reads were counted in a 2-kb window centred around all transcription start sites without further normalization.

Enhancers were classified according to the definitions previously reported (*Creyghton et al., 2010*). In brief, active enhancer regions required a significant enrichment for both, H3K4me1 and H3K27ac, while inactive enhancers were defined as regions containing only H3K4me1 (*Creyghton et al., 2010*). Total H3 ChIP-seq data to generate the meta plots around the TSSs and to calculate enrichments were extracted from *Mikkelsen et al., 2007*. Read densities ± 15 kb around TSSs were binned in 1 kb windows and plotted as mildly smoothed lines using R (lowess, f = 0.05).

## Western blotting analysis and protein pull-down

For histone extraction, nuclei were isolated by incubating the cell pellets 10 min in TEB buffer (0.5% Triton X-100 in PBS) and histones were acid extracted with 0.2 N HCl overnight at 4°C. After extraction, protein concentration was measured according to the Bradford method and equal amounts of protein were separated on SDS-PAGE and transferred onto nitrocellulose membrane.

For nuclear extracts preparation, HeLa cell pellets were incubated with TEB buffer as before, but then nuclei were incubated in extraction buffer (0.3% SDS, 0.8 M NaCl, 0.3% NP40) diluted in water to a final NaCl concentration of 0.4 M and sonicated.

For H3 variants quantification, histones were acid extracted as described above from HEK293 cell lines stably expressing different FlagHA-H3 variants and immunoprecipitated using anti-Flag beads (Sigma-Aldrich). Bound FlagHA-tagged H3s were eluted by boiling in Leammli buffer. Quantification was done using the ImageJ software and the ratio between specific H3K64ac and HA signals (loading control) was calculated.

For Nap1 pull-down, His-tagged Nap1 was pre-incubated with different histone species in incubation buffer (10 mM Tris–HCl pH 8.0, 100 mM NaCl, 100 ng/µl BSA, 0.05% NP-40, 1 mM EDTA, 1 mM DTT, 5% glycerol) for 30 min at 30°C 300 RPM. Samples were then supplied with nickel beads (His-Select Nickel Affinity Gel, Sigma) and incubated for 4 hr on the wheel at 4°C. Bound material was washed with wash buffer (10 mM Tris-HCl pH 8, 250 mM NaCl, 0.1% NP-40, 1 mM EDTA, 1 mM DTT, 5% glycerol) and elution was achieved by boiling the beads in Leammli buffer.

## Overexpression and siRNA knock-downs

For HATs overexpression, a tagged p300 HAT domain was cloned in pcDNA3 vector and 5 µg of plasmid were transfected together with a carrier DNA plasmid (0.5 µg) expressing mCherry. HEK293 cells were grown on poly-L-lysine-coated coverslips, transfected with the conventional calcium-phosphate method and processed for immunofluorescence (see below for details) after 24 hr.

For HATs knock-down RNAiMAX (Invitrogen, Carlsbad, CA) was used and a reverse transfection protocol was applied following the supplier's instructions. The siRNAs used were an ON-target plus SMART pool from Dharmacon (Thermo Fisher Scientific Inc.): p300, L-003486-00-0010; CBP, L-003477-00-0010; GCN5, L-009722-00; PCAF, L-005055-00; scramble negative control, D-001810-10. hMof siRNA was purchased from MWG (Eurofins, Germany) with sequence 5′-GUGAUCGAGUCUCGA GUGArUrU-3′. Briefly, a mixture of 50 nM total of siRNA and lipofectamine were prepared in the wells and subsequently MCF7 cells were plated on it and grown for 72 hr.

## HAT assay

Full-length HATs were expressed and purified from Sf9 cells as described elsewhere (*Kraus and Kadonaga, 1998*). Recombinant GST-tagged HAT domains were expressed in *E. coli* and then purified. For the HAT assay, recombinant histones H3 (wt and K64A) and H4 were incubated with the specific HAT in the presence of acetyl-CoA (without in case of mock reaction) in HAT buffer (50 mM Tris–HCl pH8.0, 7% glycerol, 25 mM NaCl, 0.1 mM EDTA, 5 mM DTT) for 1 hr at 30°C. The reaction was then analysed by western blotting with the specific H3K64ac antibody and the activity of the enzyme was checked using antibodies against known specific targets.

## Antibodies

Histone modifications primary antibodies: H3K4me3 (Millipore, Billerica, MA), H3K9ac (Cell Signaling, Boston, MA), ChIP grade H3K9ac (Abcam, UK), H3K9ac (Millipore, Billerica, MA), H3K9me3 (Millipore), H3K14ac (Millipore), H3K18ac (Cell Signaling), H3K18ac (Abcam), ChIP grade H3K27ac (Abcam), H3K27me3 (Millipore), H3 (Abcam), H4K16ac (Santa Cruz Biotechnology Inc., Dallas, Texas).

Non-histone primary antibodies: HA (Abcam).

## Immunofluorescence (IF) staining

Cells were fixed in 4% paraformaldehyde/2% sucrose for 15 min, washed three times in cold PBS and permeabilised with 0.5% Triton X-100 in PBS for 20 min. After washing in PBS, cells were blocked in 3% BSA/PBS and then stained with specific primary antibodies overnight at 4°C. Fluorescent secondary antibodies (FITC or RedX from Jackson Immunoresearch, UK) were then used for detection. Coverslips were either directly mounted with Vectashield containing DAPI (Vector laboratories, UK) or stained with DAPI for 10 min and then mounted with DABCO (Sigma-Aldrich).

Confocal microscopy was performed in a Leica TCS SP2/MP inverted confocal microscope using a 63× or 100× oil objective. Images were acquired along the Z-axis every 0.5 µm.

## Immunofluorescence on tubules

Tubules of mice were prepared as previously reported (*Kotaja et al., 2004*). Cells were fixed in 90% ethanol for 10 min at RT. They were permeabilised with 0.2% Triton X-100 and 0.5% saponine in PBS for 15 min at RT and blocked with 5% milk in PBS. They were incubated with anti-H3K64me3 at 4°C overnight, followed by incubation with Alexa488 conjugated secondary antibody for 30 min at 37°C. Finally, cells were counterstained with DAPI and mounted in Vectashield with DAPI (Vector laboratories).

## Immunofluorescence on testis imprint

Imprints of mice testes were prepared and fixed in 90% ethanol for 10 min at RT. They were permeabilised with 0.6% Triton X-100 in PBS containing for 15 min at room temperature and blocked with 4% BSA and 0.2% Tween in PBS. They were incubated with anti-H3K64ac at 4°C overnight, followed by incubation with Alexa488 conjugated secondary antibody for 30 min at 37°C.

## Immunohistochemistry (IHC) of testis cross-sections

AFA's fixed, paraffin-embedded testicular sections (7 µm) were deparaffinised using toluene and hydrated through graded series of ethanol (100%, 90%, 70%). The sections were then incubated with proteinase K (20 µg/ml) in PBS containing 0.5% SDS for antigen retrieval, 15 min at 37°C. They were then blocked by incubation with PBS containing 5% milk for 30 min. Immunostaining was performed by three sequential experimental steps: incubation with anti-H3K64ac antibody overnight at 4°C, washed, and then incubated with a biotinylated secondary antibody for 30 min at RT. The slides were washed in PBS with 0.5% milk; the final detection was performed using the ABC Elite kit Vectastain and the DAB peroxidase Substrate kit (Vector Laboratories) according to the manufacturer's instructions. The slides were washed and stained with PAS (Periodic Acid Schiff; Sigma-Aldrich) and hematoxylin eosin, dehydrated, and mounted in Eukitt (Sigma-Aldrich). The testis tubules sections were observed with a transmission light microscope and staged according to the criteria previously described (*Russel et al., 1990*).

## Expression and purification of K64 acetylated H3

BL21 gold cells were transformed with plasmid pAcKRS-3 encoding for the tRNA acetyl-lysine synthetase and pCDF PylT-1, and containing the amber suppressor tRNA and the H3 cDNA wild type, K64AMBER, or K9AMBER codon. The cells were then cultured as described before (*Neumann et al., 2009*). To purify the recombinant histones, cells were resuspended in wash buffer (50 mM Tris-HCl pH7.5, 100 mM NaCl, 1 mM β-mercaptoethanol) supplemented with 20 mM Nicotinamide (NAM) and 10 mg/ml lysozyme and then sonicated. The inclusion bodies were washed with the same buffer supplemented with 1% Triton X-100 and then histones were extracted with unfolding buffer (6 M Guanidium HCl, 20 mM Na-acetate pH5.2, 1 mM DTT) for 1 hr at RT. The extracted proteins were dialysed against SAU200 buffer (7 M urea, 20 mM Na-acetate pH5.2, 200 mM NaCl, 5 mM β-mercaptoethanol, 1 mM EDTA) before loading the sample on a ResourceS ion exchange column (GE Healthcare Life Sciences).

## Nucleosome reconstitutions

Nucleosomes were assembled using DNA fragments derived from the MMTV-A positioning sequence (*Flaus and Richmond, 1998*; *Flaus and Owen-Hughes, 2003*; *Ferreira et al., 2007*), and were purified as unmodified or acetylated H3/H4 tetramers and a stoichiometric amount of H2A/H2B dimers. All DNA fragments were made by PCR using fluorescently labelled primers. The PCR products were purified by anion exchange chromatography on a 1.8 ml SOURCE 15Q column (GE Healthcare Life Sciences) or native (0.5× TBE) PAGE. The notation *xAy* denotes the 147-bp MMTV-A sequence with flanking DNA of *x* and *y* bp on the upstream and downstream side, respectively. Typically, assembly was performed by salt-gradient dialysis using a double-dialysis method (unless specified otherwise), as follows—reactions (25–35 µl) were placed in microdialysis buttons, which were placed inside a dialysis bag containing 30 ml 0.5× TE and 2 M NaCl, the dialysis bag was then dialysed overnight against 2 L of 0.5× TE at 4°C.

## Salt-dependent disassembly of nucleosomes monitored by quenched FRET of AlexaFluor488

The 147-bp DNA fragment 0A0 was internally labeled with the fluorescent dye AF488 and the dark quencher BHQ1. The DNA was made by PCR using primers that contained an internally labeled thymidine 35 bp from the 5′ end of the primer (indicated as bold and underlined).

Forward AF488 primer:
-ACTTGCAACAGTCCTAACATTCACCTCTTGTGTG**T**TTGTGTCT
Reverse BHQ1 primer:
-CAAAAAACTGTGCCGCAGTCGGCCGACCTGAGGG**T**CGCCGGGG

Unmodified and acetylated nucleosomes were assembled on the doubly-labeled DNA and the integrity of reconstituted nucleosomes was checked by running equal amounts of each nucleosome

reconstitution (equivalent to 600 ng DNA) on 0.5× TBE 5% native polyacrylamide gels. To measure qF the nucleosomes were diluted to 40 nM in the final dialysis buffer and then diluted with an equal volume of 2× NaCl stock solutions, yielding final NaCl concentrations from 0 M to 2.0 M. Reactions were left for at least 15 min to reach equilibrium, transferred to 96-well half-area black non-binding surface microplates (Corning, Corning NY) and AF488 fluorescence measured on a BMG Labtech FLUOstar Optima plate-reader using 488p and 520p excitation and emission filters, respectively. Titrations using DNA alone were performed and confirmed that NaCl-induced changes in qF of nucleosomes was due to nucleosome disruption and not solute-dependent effects on the fluorophore. Upper and lower plateaus for each titration were used to normalise the data on a scale from 1–0. The plateau values were calculated by fitting the raw data using the sigmoid function described below in Microcal Origin 8.6 Software. Average titration curves were calculated from normalised data from three separate titrations (using different nucleosome reconstitutions) for unmodified and acetylated nucleosomes and fit again with the sigmoid function to determine the average NaCl concentration at which the normalised qF ($qF_{norm}$) is 0.5 ($_{C0.5}$):

$$qF_{norm} = A2 + \frac{A1 - A2}{1 + e^{(C - C_{0.5})/dC}}$$

*A1* and *A2* are the lower and upper plateau values, respectively, *C* the NaCl concentration and *dC* a constant.

## Competitive nucleosome assembly experiments

The competitive reconstitution procedure is based on that described in *Thåström et al. (2004)* and *Manohar et al. (2009)*. However, we found that using the dilution method described below gave more consistent results than the double-dialysis method. Reactions were initially set up in 14.5 µl 1× TE and 2 M NaCl and contained 5 µg UltraPure sheared salmon sperm DNA (Invitrogen) as non-specific competitor, 1 µg Cy3-labelled 0A0 DNA as the specific assembly fragment, 80 pmol H2A/H2B dimer, 40 pmol of unmodified or acetylated H3/H4 tetramer. A mastermix containing all components except the tetramer was assembled, split into separate reaction tubes, and the required amount of concentration-matched unmodified/acetylated tetramer added. This was to ensure the maximum amount of consistency between the reactions. The NaCl concentration was then decreased by addition of 1× TE 0.1 mM DTT to give concentrations of 1.5, 1.0, 0.75, 0.6, and 0.4 M, with sufficient time given for the reactions to equilibrate at each NaCl concentration (15–30 min). Once reactions had reached 0.4 M NaCl they were split into two separate microdialysis buttons and dialysed against 1× TE 0.1 mM DTT for several hours to reduce the NaCl to 0 M. The reactions were split in two to assess variability during dialysis; however, we found they did not vary by more than 3%. The reactions were then mixed with sucrose to a final concentration of 10% (w/v) and run on 0.2× TBE 6% native polyacrylamide gels run at 200 V for 75 min at 4°C. Reactions using unmodified and acetylated tetramer were always set up side-by-side. The data are the average of three different batches of assembly reactions performed on different days. Band intensities were quantified in AIDA software. The average assembly efficiency of the unmodified nucleosome reactions from each batch was defined as 1.0 and used as a reference for normalisation.

## ATP-driven remodelling reactions

Nucleosomes were assembled on Cy3- or Cy5-labelled 54A18 DNA for RSC remodelling reactions or 54A0 for Chd1 reactions. Recombinant Chd1 was produced as described in *Ryan et al. (2011)*. TAP-tagged RSC was produced as described in *Ferreira et al. (2007)*. The remodelling reactions were carried out essentially as described in *Ferreira et al. (2007)*; *Somers and Owen-Hughes (2009)* and *Ryan et al. (2011)*. Briefly, the reactions contained 50 mM Tris pH 7.5, 50 mM NaCl, 3 mM MgCl$_2$, and 1 mM ATP (or 1 mM ATPγS where indicated), 50 nM each of H3K64-acetylated nucleosomes and unmodified nucleosomes on Cy3- and Cy5-labelled DNA, respectively, and either 1 nM Chd1 or 0.5 nM RSC. The reactions were incubated at 30°C and samples taken at 0, 5, 10, 15, 20, 30, and 45 min. The reactions were stopped by the addition of 500 ng HindIII digested λ DNA and sucrose to 5% w/v and placing on ice. Products were resolved on 0.2× TBE 5% native polyacrylamide gels run for 3 hr at 300 V at 4°C. Gels were imaged on an FLA-5100 fluorescence scanner. Band intensities were quantitated in AIDA software. Initial rates were obtained by fitting using the following equation in MicroCal Origin 7.0 software and solving the derivative at zero time:

$$y = a\left(1 - e^{-bx}\right)$$

where, *a* describes the value of the asymptote of the curve and *b* the relative rate in terms of normalized fraction of nucleosomes repositioned per minute. The fraction is a normalised unitless value and so the units for the rate are just $min^{-1}$.

## Single-molecule FRET (smFRET) analysis

Unmodified or H3K64 acetylated nucleosomes were assembled on the 147-bp DNA fragment 0A0, which was labeled with fluorescent dyes attached to the same thymidine of the primer sequences shown before (see also 'Salt-dependent disassembly of nucleosomes monitored by quenched FRET of AlexaFluor488' in 'Materials and methods' section), 6-Tamra (forward primer) and alexa647 (reverse primer). PCR products were purified by DNA precipitation followed by size exclusion chromatography using a Superose 6 PC 3.2/30 column (GE Healthcare Life Sciences).

DNA and histone octamers containing either unmodified H3 or H3K64ac were assembled to mononucleosomes by salt-gradient dialysis according to the double-dialysis method described above, but placing microdialysis buttons into 300 ml 0.5× TE-NP40 and 2 M NaCl, and dialysing them overnight against 3 l of 0.5× TE-NP40 and 50 mM NaCl at 4°C. Finally, the buttons were placed in 1 L 0.5× TE-NP40 buffer for 1 hr. A 1:20 mixture of labelled and unlabelled DNA was used for the assembly to generate a mixture of labelled and unlabelled nucleosomes and thus avoid nucleosome disruption due to low nucleosome concentration during the single-molecule experiments.

For smFRET measurements, nucleosomes were diluted in measurement buffer (10 mM Tris–HCl pH 7.6, 1 mM EDTA, 0.05% NP40, 0.1 mg/ml BSA and varying amounts of NaCl, ranging from 0 to 1 M) to a concentration in the order of 500 pM, corresponding to a picomolar concentration of the fluorescent species. Samples were then kept at 4°C for at least 18 hr to allow for adaptation to the respective final salt concentration. A 20 µl drop was placed on a PEG-coated coverslip and measurements were performed for 10 min at room temperature. At least three measurements were performed for each salt concentration on a custom built confocal microscope using freely diffusing molecules and time-correlated single photon counting (TCSPC), as described in *Bönisch et al. (2012)*. Photon arrival times were recorded by a Hydra Harp 400 (PicoQuant GmbH, Germany).

To select bursts of fluorescence, an all-photons burst search (APBS, *Nir et al., 2006*) was performed on the recorded data, requiring a minimum of five photons in 500 µs and at least 30 photons in total per burst. For each burst, FRET efficiency and stoichiometry were calculated from the recorded signals as explained in *Bönisch et al. (2012)*. Only bursts with more than 50 photons in total were considered for further analysis. The populations of singly-labeled molecules were excluded by a stoichiometry (*Sto*) threshold of 0.1<*Sto*<0.7 and rare multi-molecule events were excluded by limiting the time deviation signals (TDS) to TDS<1 and $TDS_{red-PIE}$<0.4 as explained in *Bönisch et al. (2012)*.

Because of the chosen position of the dyes, intact nucleosomes showed an intermediate to high FRET efficiency, while disassembled nucleosomes showed a very low FRET efficiency (dyes position shown in *Figure 5A*). Thus, two FRET populations, represented by two Gaussian distance distributions, were fitted to the data by probability distribution analysis (PDA, *Antonik et al., 2006*) assuming a Förster radius of 60 nm. Additional measurement allowed us to estimate a background signal of 0.2 kHz in all channels, 4% crosstalk, and a γ-factor of 0.7, as a correction factor for differences in the quantum yields of the dyes and the wavelength-dependent detection efficiencies of the setup. Measures of the higher FRET efficiency corresponded to the fraction of intact nucleosomes at the respective salt concentration. These values were normalized to the higher FRET efficiency value at 0 mM NaCl to allow comparison between unmodified and acetylated nucleosomes. The sigmoidal fit function described in the section 'Salt-dependent disassembly of nucleosomes monitored by quenched fluorescence (qF) of AlexaFluor488 (AF488)' was used to fit the data and retrieve $C_{0.5}$.

## Histone eviction assay

The histone eviction assay was adapted from a protocol developed by *Ito et al. (2000)* and *Sharma and Nyborg (2008)*. Chromatin templates containing unmodified H3, H3K64ac, or H3K64R were assembled on an ~680-bp linear DNA fragment labeled with biotin on the 5′ end. Chromatin was assembled using the salt dilution method as described in *Gutierrez et al. (2007)*, where 20 µl reactions at 1 M NaCl were diluted with 1× TE, 1 mM DTT to 0.8, 0.6, 0.4, 0.2, and 0.1 M (final dilution buffer 2×: 2× TE, 20% glycerol, 0.1% NP40, 200 ng/µl BSA, 2 mM DTT, 1 mM Na-butyrate, protease inhibitors). Each dilution step was carried out at 4°C, 400 RPM, for 45 min. Assembled chromatin was analysed on native 5% polyacrylamide gel run in 0.2× TBE for 90 min at 150 V. Chromatin templates

were bound on streptavidin beads for 3 hr at 4°C (10 mM Tris–HCl pH8, 100 ng/µl BSA, 100 mM NaCl, 0.05% NP-40, 1 mM EDTA, 1 mM DTT, 0.1 mM AEBSF, 0.5 mM Na-butyrate, 10% glycerol) and the immobilised template was incubated (10 mM Tris–HCl pH 8.0, 100 ng/µl BSA, 100 mM NaCl, 0.05% NP-40, 1 mM EDTA, 1 mM DTT, 1 mM Na-butyrate, 5% glycerol) first with Gal4-VP16 for 20 min at 30°C, 450 RPM, and then with full-length p300 and Nap1 (Drosophila, full length, expressed in Sf9 cells) for 15 min. Then, a mixture of radioactive $^3$H-acetyl-CoA and 100 µM cold acetyl-CoA was added to the samples and incubated for 40 min at 30°C. Before this last incubation, each reaction was supplemented with 1 µg of a supercoiled plasmid, on which the evicted octamers are assembled by Nap1. The supernatant fraction was recovered and assayed in western blot for total H3 amounts. The bound fraction was washed three times and analysed on SDS-PAGE for p300 specific activity onto chromatin.

## TPA induction and qPCR analysis

NIH3T3 cell lines stably expressing different FlagHA-H3.3 species were seeded in selection medium containing 10% FBS and then starved for 2 days in starvation medium without FBS. Gene induction was stimulated supplementing the cells with fresh starvation medium supplemented with 25 nM TPA (Cell Signaling). After the indicated times of induction, total RNA was isolated using Quick RNA Miniprep kit (Zymo Research, Irvine, CA). Equal amounts of RNA were employed for first strand cDNA synthesis using RevertAid H Minus First Strand cDNA Synthesis Kit (Thermo Fisher Scientific). cDNA samples were then used as templates to perform quantitative real-time PCR using SybrGreen (Thermo Fisher Scientific) on LightCycler 480 II (Roche). Fold change values in gene expression were calculated relative to the untreated samples for each cell line using the—dCt method. Primers used for qPCR are listed in *Supplementary file 1*.

## Acknowledgements

We thank A Izzo and all members of RS laboratory. We are grateful to H Neumann and R Margueron for providing expression plasmids and helpful advices. We acknowledge C Beisel and I Nissen for handling samples, M Stadler, D Gaidatzis, L Burger and R Ivanek for providing a computational framework for data analysis. We are grateful to P Eberling for peptide synthesis.

## Additional information

### Funding

| Funder | Grant reference number | Author |
| --- | --- | --- |
| Deutsche Forschungs Gemeinschaft | SFB 746 | Robert Schneider |
| Fondation pour la Recherche Medicale | | Robert Schneider |
| European Research Council (ERC) | Protmod | Robert Schneider |
| Wellcome Trust | 95062 | Tom Owen-Hughes |
| Max Planck Society | | Gerhard Mittler, Robert Schneider |
| Agence Nationale de Recherche | CoreAc | Robert Schneider |
| NHMRC Australia | 457137 | Daniel P Ryan |
| Cancer Research UK | Program Grant | Andrew J Bannister |
| ANR France | EpiSperm2 | Saadi Khochbin |
| INCa | | Saadi Khochbin |
| Ligue contre le Cancer | Equipe labellisée | Robert Schneider |

The funders had no role in study design, data collection and interpretation, or the decision to submit the work for publication.

### Author contributions

VDC, SD, Conception and design, Acquisition of data, Analysis and interpretation of data, Drafting or revising the article; FM, DPR, GM, Acquisition of data, Analysis and interpretation of data, Drafting or revising the article; EM, PT, FMR, Acquisition of data, Analysis and interpretation of data; SK, EK,

MH, LH, AF, Conception and design, Acquisition of data, Analysis and interpretation of data; AJB, Conception and design, Drafting or revising the article; JM, SK, RF, DS, Analysis and interpretation of data, Drafting or revising the article; TO-H, RS, Conception and design, Analysis and interpretation of data, Drafting or revising the article

## Ethics

Animal experimentation: For this study animals were euthanized following a procedure approved by ad hoc committees of the University Joseph Fourier and all of the investigators directly involved. All investigators have an official animal-handling authorization obtained after 2 weeks of intensive training and a final formal evaluation. Effort was made to minimize suffering.

## Additional files

### Supplementary files

• Supplementary file 1. Oligonucleotides used in this study.

### Major dataset

The following dataset was generated:

| Author(s) | Year | Dataset title | Dataset ID and/or URL | Database, license, and accessibility information |
|---|---|---|---|---|
| Di Cerbo, et al. | 2014 | Acetylation of histone H3 at lysine 64 regulates nucleosome dynamics and facilitates transcription | GSE35355; http://www.ncbi.nlm.nih.gov/geo/query/acc.cgi?acc=GSE35355 | Publicly available at the Gene Expression Omnibus (http://www.ncbi.nlm.nih.gov/geo/). |

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
