## [Decision Letter]

Thank you for sending your work entitled “Acetylation of histone H3 at lysine 64 regulates nucleosome dynamics and facilitates transcription” for consideration at *eLife*. Your article has been favorably evaluated by a Senior editor and 3 reviewers, one of whom is a member of our Board of Reviewing Editors.

The Reviewing editor and the other reviewers discussed their comments before we reached this decision, and the Reviewing editor has assembled the following comments to help you prepare a revised submission.

In this study, the authors identified and characterized a novel histone acetylation site at H3K64. This residue, located on the outer (lateral) surface of the histone octamer, is close to the DNA within the nucleosome structure and is thus potentially interesting because its acetylation may disrupt the nucleosome in a manner similar to that caused by H3K56ac. H3K64ac was identified by mass spec and found to be present in mice and humans (Figure 1). Genome-wide analyses revealed that H3K64ac correlated with enhancers as well as active promoters (Figures 2 and 3). Acetylation of H3K64 appears to be performed largely, but not necessarily exclusively, by p300/CBP (Figure 4). Acetylation of H3K64 may cause a slight destabilization of nucleosomes at NaCl concentrations greater than 0.35 M (Figure 5). The localization of H3K64ac during spermatogenesis was also examined (Figure 5). Figure 6 addresses, somewhat incompletely, the possible role of H3K64ac in histone eviction and transcription.

Hence, this study describes acetylation of H3K64 in mice and humans and finds that p300/CBP largely contribute to this acetylation. In a manner similar to other acetylation marks in H3, H3K64ac is found in transcriptionally active chromatin. H3K64ac may destabilize nucleosome stability and facilitate transcription. The main strengths of this work are the identification of H3K64ac, the prominent role of p300/CBP in acetylating H3K64, and the association of H3K64ac with active chromatin. Two key concerns are the specificity of the H3K64ac antibodies and the abundance of H3K64ac relative to H3K18ac and H3K27ac, which are also generated by p300/CBP. The functional studies are also less convincing.

Major comments:

1) The authors need to test more thoroughly the specificity of the H3K64ac antibodies, as many results depend upon whether or not the antibodies are specific. In particular, the authors need to test whether singly acetylated peptides containing H3K18ac or H3K27ac (sites that are also acetylated by p300/CBP) are recognized by the H3K64ac antisera. Thus, both Figure 1 and Figure 1—figure supplement 2 should be expanded to test these peptides.

2) It is essential to determine the relative abundance (by mass spec) of H3K64ac compared to H3K18ac and H3K27ac, as this issue is relevant to the possible roles of H3K64ac in the cell.

3) In addition, if the abundance of H3K64ac relative to H3K18ac or H3K27ac is very low, then the specificity of the antibodies (point 1 above) becomes even more critical. In this scenario, the authors should make the H3K64R mutation in the epitope tagged H3.1 and H3.3 histones that they use in later figures, express it in cells and show by western blot that their antibody no longer recognizes the tagged H3 from cells to prove specificity of the antibody. This is especially important because CBP is the apparent HAT for H3K64ac and the pattern of localization of the H3K64ac antibodies to promoters is similar to that of other H3 acetylation marks.

4) Figure 2. The meta gene analysis in Figure 2 shows a dramatic dip right over the TSS and a peak on either side for H3K64ac signal normalized by Input. It is well-known that there is histone displacement at the TSS to proximal regions to allow for the regulatory machinery to bind. Therefore, the meta gene results could be indicative of higher H3 concentration in the proximal regions of the TSS (and corresponding depletion over TSS) rather than a specific increase in H3K64ac. The authors should normalize the signal ChIP to H3 rather than input for this particular case.

5) Figure 5. The experiments on H3K64ac being involved in histone eviction are problematic. First, the line is a poor fit to the data. The actual data points indicate that there is, at best, roughly a 50 mM difference in the NaCl concentration between the H3K64ac data and the other data.

6) Figure 6 is incomplete. For each different histone (unmod, K64R, K64ac), data needs to be shown for +/- NAP1, +/- Gal4-VP16, +/- p300, and +/- AcCoA. It is not possible to interpret the data in the absence of the missing conditions. Also, does the supernatant contain nucleosomes as shown in the Figure? If so, do the nucleosomes have NAP1 bound to them as shown in the Figure? In this experiment, is it simply possible that NAP1 cannot recognize H3K64R?

7) The data in Figure 6 are interpreted as showing that a K to Q mimic of lysine 64 increases transcription. However, the increased transcription in the K64Q mutant is subtle (less than 2 fold) and is only seen at some genes and at some time points. The interpretation of the results needs to reflect the data more accurately.

[Editors' note: further clarifications were requested prior to acceptance, as described below.]

1) Can the authors indicate the source/origin (yeast, human, etc.) of the Nap1?

2) Figure 5. The lines still do not seem to be a good fit to the data, especially the H3K9ac line.

3) Figure 5. One additional complication in the interpretation of the data is that H2A-H2B begin to dissociate from chromatin at around 0.6 M NaCl.

---

## [Author Response]

*1) The authors need to test more thoroughly the specificity of the H3K64ac antibodies, as many results depend upon whether or not the antibodies are specific. In particular, the authors need to test whether singly acetylated peptides containing H3K18ac or H3K27ac (sites that are also acetylated by p300/CBP) are recognized by the H3K64ac antisera. Thus, both*
Figure 1
*and*
Figure 1—figure supplement 2
*should be expanded to test these peptides*.

We agree with the reviewer that testing the H3K64ac antibody for recognition of other p300/CBP acetylation sites is an important control. To this end we:

A) Synthesized H3K18ac and H3K27ac peptides and included these peptides in our specificity screening. As shown in new Figure 1—figure supplement 2 we found that apparently our serum recognises H3K64ac with an affinity significantly greater than the other acetylated peptides.

B) Have additionally included H3K18ac and H3K27ac peptides in our peptide competition assays. In this assay we observe complete competition with the H3K64ac peptide. In stark contrast, we observe no competition using the H3K18ac or H3K27ac peptides (new Figure 1—figure supplement 2). Together, these data provide strong evidence that the H3K64ac antibody preparation is indeed highly selective.

*2) It is essential to determine the relative abundance (by mass spec) of H3K64ac compared to H3K18ac and H3K27ac, as this issue is relevant to the possible roles of H3K64ac in the cell*.

To address this point we performed quantitative mass spec analysis of H3K18ac, H3K27ac and H3K64ac. We find a relative abundance of ∼200:2:1 for these three acetylation sites. We would like to point out that the rather unexpected ratio of H3K18ac to H3K27ac (200:2) is in line with that reported recently in a systematic mass spectrometrical study (LeRoy et al. 2013) and that the abundance of H3K64ac and H3K27ac we found is comparable. H3K27ac is an established signature for active regulatory elements (and to some extent also active genes) and we show that H3K64ac is specifically marking active transcription start sites. These findings fit well with the similar abundances of these two acetylations.

We would like to emphasise that the selectivity of the H3K64ac anti-serum for H3K64ac in our assays is – compared to H3K18ac and H3K27ac – higher than the ratio of their relative abundances (Figure 1—figure supplement 2). Furthermore upon mutation of H3K64 to an R we detect a loss of H3 recognition by the H3K64ac antibody on all three H3 variants (Figure 3—figure supplement 1).

*3) In addition, if the abundance of H3K64ac relative to H3K18ac or H3K27ac is very low, then the specificity of the antibodies (point 1 above) becomes even more critical. In this scenario, the authors should make the H3K64R mutation in the epitope tagged H3.1 and H3.3 histones that they use in later figures, express it in cells and show by western blot that their antibody no longer recognizes the tagged H3 from cells to prove specificity of the antibody. This is especially important because CBP is the apparent HAT for H3K64ac and the pattern of localization of the H3K64ac antibodies to promoters is similar to that of other H3 acetylation marks*.

We originally used H3.2 as an antibody control because we detect lower levels of H3K64ac on this subtype and therefore we considered it as a very stringent antibody control. However, as suggested by the reviewers we have now performed the same test with the H3.1 and H3.3 variants expressed in vivo. As shown in new Figure 3—figure supplement 1, mutations of H3K64 to R in H3.1 and H3.3 result (as shown previously for H3.2) in a loss of H3K64ac detection by our antibody.

To further validate the specificity of the H3K64ac antibody we have now subjected native nucleosomes to tryptic digestion followed by Western blotting. This mild digestion removes the histones tails and leaves the H3 core region (protected by the DNA) predominantly intact. As shown in new Figure 1—figure supplement 2, this results, as expected, in a total loss of H3K9ac, H3K18ac and H3K27ac signals. However, the tailless H3 band is still recognised by the H3K64ac antibody with a comparable intensity to the intact and undigested H3, strongly arguing against cross reactivity towards acetylation sites in the H3 tail (and indeed any other acetylated sites within the H3 N-terminal tail).

Together these additional experiments provide strong experimental support for the specificity of the H3K64ac antibody we used.

*4)*
Figure 2*. The meta gene analysis in*
Figure 2
*shows a dramatic dip right over the TSS and a peak on either side for H3K64ac signal normalized by Input. It is well-known that there is histone displacement at the TSS to proximal regions to allow for the regulatory machinery to bind. Therefore, the meta gene results could be indicative of higher H3 concentration in the proximal regions of the TSS (and corresponding depletion over TSS) rather than a specific increase in H3K64ac. The authors should normalize the signal ChIP to H3 rather than input for this particular case*.

The finding of a dip in enrichments over the TSS has been often reported before, e.g., it has been consistently found in meta profiles for most active histone marks as well as Pol II. As the reviewer points out, this dip corresponds to the so-called ‘nucleosome-free region’ (NFR). The possibility that the proximal enrichment of H3K64ac could be due to highly increased H3 concentrations, appears unlikely as to our knowledge this has not been reported for mouse stem cells. Instead, it has been repeatedly shown that histone density up and downstream of the TSS of active genes is rather uniform in ES cells (Teif et al. 2012, Kelly et al. 2012).

Nevertheless, we acknowledge that there is an ongoing discussion in the field about the normalisation of ChIP data to input versus normalisation to H3. To address the referee’s concerns we have now also plotted the H3 enrichments in a similar meta analysis as in Figure 2. As shown in the new Figure 2—figure supplement 1 H3 is only slightly enriched proximal to the NFR and therefore the strong enrichment of H3K64ac cannot be explained by higher H3 concentration. We have included this analysis and clarified the corresponding text in the revised manuscript (main text and corresponding methods and figure legends of supplementary information). We hope that this new analysis addresses now the referees’ concern.

*5)*
Figure 5*. The experiments on H3K64ac being involved in histone eviction are problematic. First, the line is a poor fit to the data. The actual data points indicate that there is, at best, roughly a 50 mM difference in the NaCl concentration between the H3K64ac data and the other data*.

A) To address the referees concern regarding the fit of the salt-induced nucleosome disassembly data, we have explored alternative modeling algorithms and re-analysed the data using a different sigmoid model that generates a 'better' fit across the three different datasets (R^2^ values all above 0.99). This new analysis is now used in Figure 5 and the new line fits better to the K64ac data. It has resulted in some minor changes in the *C*_*0.5*_ values, but these are still consistent with our previous analysis/interpretation. The values are as follows (new model vs old model): unmodified – 697 ± 9 mM vs 690 ± 11 mM; H3K9ac – 691 ± 13 mM vs 682 ± 12 mM; H3K64ac – 622 ± 16 mM vs 605 ± 12 mM. We have now included this in a new model in Figure 5.

B) With respect to the magnitude of the change between H3K64ac and unmodified/H3K9ac we strongly believe that the 60-70 mM difference is meaningful and significant, since it is consistent with previously published studies using similar methodologies. [18] showed that simultaneous acetylation of all sites in H3 results in a decrease in nucleosome salt stability of ∼100-130 mM NaCl. Furthermore, [39] showed that a fundamental change in the histone content of the nucleosome, the incorporation of the histone H2A variant H2A.Z, increases salt stability up to ∼70 mM. It is well documented that H2A.Z significantly increases the stability/compaction of nucleosome arrays (Fan et al. 2002) and reduces transcription of chromatin templates in vitro (Zhou et al. 2007). Thus, the decrease in salt stability we observe for H3K64ac appears to be in a range consistent with real changes in nucleosome dynamics. This is also supported by our additional in vitro and in vivo data. Notably, not all core histone acetylations result in destabilised nucleosomes in vitro. For instance, [35] reported that H3K56ac nucleosomes show no significant difference in salt stability relative to unmodified nucleosomes. Thus, the actual site of acetylation within the nucleosome seems important for its destabilisation rather than acetylation per se. We have expanded this explanation in the discussion section of our revised manuscript.

Notwithstanding the above, the results from the quoted studies should only be used as a relative guide and they should not be used to make absolute comparisons, as each study uses different fluorophore combinations, different dye-attachment sites, different instrumental setups, and different DNAs. Nevertheless, they consistently describe comparable changes in salt stability that can affect chromatin function.

C) To experimentally address the referees’ concern we have now performed additional stability measurements, using a more sensitive method, single-molecule FRET analysis. These new data are now included as Figure 5—figure supplement 4 and they confirm, at a single-molecule level, our original data that the presence of H3K64ac decreases nucleosome stability.

We hope that with these new data and the detailed explanations the reviewer will be now convinced that the differences that we report are consistent and meaningful.

*6)*
Figure 6
*is incomplete. For each different histone (unmod, K64R, K64ac), data needs to be shown for +/- NAP1, +/- Gal4-VP16, +/- p300, and +/- AcCoA. It is not possible to interpret the data in the absence of the missing conditions. Also, does the supernatant contain nucleosomes as shown in the Figure? If so, do the nucleosomes have NAP1 bound to them as shown in the Figure? In this experiment, is it simply possible that NAP1 cannot recognize H3K64R*?

We agree with the referee that the original version was difficult to interpret and we apologise for this. Now, as requested, we have extended our analysis of the in vitro histone eviction. We have:

A) Included a Nap1 – control. As shown in new Figure 6—figure supplement 2, the presence of the chaperone Nap1 is required to detect histones in the supernatant. In fact, Nap1 helps to transfer the evicted histones onto a supercoiled plasmid DNA template and prevents the re-assembly of evicted histones onto the linear chromatin templates.

B) As suggested we have also included a - AcCoA control. As shown previously eviction capability in vitro is driven by acetylation (25, 45). In the absence of AcCoA, p300 activity is impaired, and this leads to strong effects on histone eviction (Figure 6). However, the pre-acetylated H3K64ac nucleosomes, due to their intrinsic higher instability are prone to easier eviction, which may occur to a low level even in the absence of AcCoA, supporting our model of destabilisation of nucleosomes by H3K64ac.

Since VP16 is required for the recruitment of p300 itself the absence of p300 and VP16 would simply mimic the absence of the essential co-factor AcCoA. Therefore, we believe that these conditions are already covered by the new -AcCoA (no acetylation) control and would not add any further information to the mechanistic effect of H3K64ac nucleosome eviction.

Altogether, the new revised figure reinforces our mechanistic model, whereby destabilisation of the chromatin template due to the presence of H3K64ac facilitates histone eviction. In agreement with this, the K64R mutation strongly impairs histone eviction in vitro. These data are now included in new Figure 6 and Figure 6—figure supplement 2. We believe that these new data firmly support our mechanistic model.

C) Following the reviewers’ comments, we realised that our schematic of the eviction assay shown in Figure 6 might have been misleading. Nap1 is known to bind both H2A-H2B dimers and H3-H4 tetramers in vitro (Andrews et al. 2008). However, we have no evidence for full nucleosome formation upon/during eviction and the conclusion from this experiment is solely that H3K64ac promotes eviction of histone H3. To avoid confusion we have adapted the model accordingly, and now the scheme shows Nap1 binding independently the H3-H4 tetramer and H2A-H2B dimers. These tetramers and dimers can then be sequestered by Nap1 onto the supercoiled plasmid DNA in the supernatant.

D) The reviewer raised an important question concerning the ability of Nap1 to bind equally to all H3 types used in the assay. To address the effect of the H3K64R mutation on Nap1 binding we performed direct in vitro interaction assays. As shown in new Figure 6—figure supplement 2, Nap1 binds to the H3K64R mutant similarly to unmodified H3 or H3K64ac. This strongly suggests that neither the H3K64R mutation nor H3K64ac affects Nap1 binding.

*7) The data in*
Figure 6
*are interpreted as showing that a K to Q mimic of lysine 64 increases transcription. However, the increased transcription in the K64Q mutant is subtle (less than 2-fold) and is only seen at some genes and at some time points. The interpretation of the results needs to reflect the data more accurately*.

As suggested by the editor we reworded the corresponding sections in the results and discussion section to make it clear that we detect changes of expression at certain genes. To substantiate the original data we have now included additional time points during gene induction. These demonstrate an increase in transcription of *c-fos* and *Egr1* genes, and to a lesser extent of *c-myc* gene, across the complete gene induction time-course. Additionally, we included now an H3K64R mutant. We also show by Western Blot analysis that the levels of the exogenously expressed histones in this system are equal.

For us this experiment demonstrates a reproducible effect of a mutation of H3K64 on gene expression. We do not know how general the effect is, however we consider a greater than 2-fold effect due to a single histone mutation as significant. This 2-fold difference has been calculated relative to wt H3, which may still be acetylated at K64. We have now also included the transcriptional effects of the H3K64R mutant, which cannot be acetylated at K64. As expected the K64R mutant has a negative effect on transcription of the analysed genes and the fold difference in transcriptional levels between K64Q and K64R is, as expected, higher than the difference between K64Q and wt H3 (up to more than 3-fold). Furthermore, we reason that a two-fold difference in transcriptional levels is already known to be meaningful in a biological context. For example, in dosage compensation where it contributes to sex determination.

[Editors' note: further clarifications were requested prior to acceptance, as described below.]

*1) Can the authors indicate the source/origin (yeast, human, etc.) of the Nap1*?

We apologize for this omission. We state now in the Material and Methods that the Nap1 used is full length Drosophila Nap1.

*2)*
Figure 5*. The lines still do not seem to be a good fit to the data, especially the H3K9ac line*.

For the original fits we had used the default weighting option in our fitting software (Origin 8.6), which gives more weight to data points with lower errors. This is the reason why the fits did not pass directly through some of the data points with larger errors bars. To address the reviewers’ concern we have removed all weighting and the lines now pass through the more variable data points. In line with this we also updated the C_0.5_values. Note that there is no change to the conclusions drawn from this experiment, i.e., the salt-dependent disruption profile of H3K64ac nucleosomes is significantly different from that of unmodified/H3K9ac nucleosomes. We hope that this addresses the concerns of the referee.

*3)*
Figure 5*. One additional complication in the interpretation of the data is that H2A-H2B begin to dissociate from chromatin at around 0.6 M NaCl*.

The important point of our results is that H3K64 acetylation sensitises nucleosomes to salt. At present, we do not fully understand how this is manifested but, given the proximity of K64 to DNA, we believe it is likely to involve perturbations of the interactions between H3 and the DNA. Nevertheless, the reviewer is correct to indicate that H2A/H2B dimers dissociate from nucleosomes at around 0.6 M NaCl. We are happy to consider that this event may play some role in the H3K64ac effect that we observe. Consequently, we now suggest that there are at least three ways by which H3K64 acetylation could affect the reduced FRET signal observed in the salt dependent dissociation assay.

i) Dissociation of DNA from the surface of the histone octamer is a prerequisite for histone dimer loss. As a result, it is possible that the loss of histone-DNA contacts caused by H3K64 acetylation (Figure 5—figure supplement 5) act to facilitate loss of histone dimers and as a consequence the observed FRET signal.

ii) Following loss of histone dimers, H3K64 acetylation may favor dissociation of DNA from histones H3 and H4.

iii) H3K64 acetylation may affect water-mediated hydrogen bonds that directly influence association of histone dimers (Figure 5—figure supplement 5). These modes of action are not mutually exclusive and all are consistent with the observed effects of H3K64 acetylation in chromatin assembly in vitro (Figure 5). They also have the potential to influence chromatin organization during cycles of assembly/disassembly at gene regulatory elements.

Additional discussion of this point has been added to the Discussion section of the revised manuscript as a new paragraph.